# *Orthobunyavirus* spike architecture and recognition by neutralizing antibodies

Jan Hellert [1], Andrea Aebischer[2], Kerstin Wernike [2], Ahmed Haouz [3], Emiliana Brocchi[4], Sven Reiche [5], Pablo Guardado-Calvo [1], Martin Beer[2] & Félix A. Rey [1]

Orthobunyaviruses (OBVs) form a distinct genus of arthropod-borne bunyaviruses that can cause severe disease upon zoonotic transmission to humans. Antigenic drift or genome segment re-assortment have in the past resulted in new pathogenic OBVs, making them potential candidates for causing emerging zoonoses in the future. Low-resolution electron cryo-tomography studies have shown that OBV particles feature prominent trimeric spikes, but their molecular organization remained unknown. Here we report X-ray crystallography studies of four different OBVs showing that the spikes are formed by an N-terminal extension of the fusion glycoprotein Gc. Using Schmallenberg virus, a recently emerged OBV, we also show that the projecting spike is the major target of the neutralizing antibody response, and provide X-ray structures in complex with two protecting antibodies. We further show that immunization of mice with the spike domains elicits virtually sterilizing immunity, providing fundamental knowledge essential in the preparation for potential newly emerging OBV zoonoses.

[1] Structural Virology Unit, Virology Department, Institut Pasteur, CNRS UMR 3569, 25-28 rue du Dr. Roux, 75015 Paris, France. [2] Institute of Diagnostic Virology, Friedrich-Loeffler-Institut, Südufer 10, 17493 Greifswald, Insel Riems, Germany. [3] Crystallography Platform, Department of Structural Biology and Chemistry, Institut Pasteur, CNRS UMR 3528, 25-28 rue du Dr. Roux, 75015 Paris, France. [4] Istituto Zooprofilattico Sperimentale della Lombardia e dell'Emilia Romagna, Via Bianchi 7, 25125 Brescia, Italy. [5] Department of Experimental Animal Facilities and Biorisk Management, Friedrich-Loeffler-Institut, Südufer 10, 17493 Greifswald, Insel Riems, Germany. These authors contributed equally: Jan Hellert, Andrea Aebischer. Correspondence and requests for materials should be addressed to M.B. (email: martin.beer@fli.de) or to F.A.R. (email: felix.rey@pasteur.fr)

OBVs are poorly studied emerging and re-emerging arthropod-borne viruses transmitted mainly by infected mosquitoes or midges[1], causing severe disease in humans and in farm animals. A striking example was the recent appearance of Schmallenberg virus (SBV) in Europe[2]. SBV first infected domestic ruminants in the summer of 2011, breaching the placental barrier in pregnant cattle and ewes to target the fetal central nervous system[3], with catastrophic consequences for the offspring[4,5]. Recurrent human pathogenic OBVs include Oropouche virus (OROV), which causes acute febrile illness in the Amazon region of South America[6], and La Crosse virus (LACV), a common cause of mosquito-borne pediatric encephalitis in North America[7].

The OBV genus belongs to the *Peribunyaviridae* family, one of the 10 virus families composing the *Bunyavirales* order. Like many other bunyaviruses, OBVs have a genome consisting of three segments of single-stranded RNA with negative polarity, termed large (*L*), medium (*M*), and small (*S*) segments. This genome segmentation opens the possibility of reassortment, i.e., the exchange of segments between two parental viruses to generate a new virus with potentially altered pathogenic properties, as is well documented in the appearance of pandemic strains of influenza viruses[8]. Emergence of natural OBV reassortants has also been well characterized[9], featuring especially exchanges of the *M* segment, which codes for the viral envelope glycoproteins. Importantly, reassorted OBVs have been implicated in severe hemorrhagic fever outbreaks in humans, as was the case of Ngari virus in Kenya and Somalia in 1997 and 1998[10].

The OBV *M* segment encodes two glycoproteins, Gn and Gc, derived from a single polyprotein precursor, with Gn encoded upstream of Gc. A non-structural protein denoted NSm is encoded in between Gn and Gc. Glycoprotein Gc is predicted to belong to the class II of membrane fusion proteins[11], i.e., to be a distant homolog of the fusion glycoproteins of the flaviviruses and the alphaviruses[12]. Gn and Gc associate cotranslationally in the ER lumen of infected cells, and the resulting heterodimer is transported to the Golgi apparatus, where new virions bud[13]. Compared with other bunyaviruses, OBV Gn is relatively small, with an ectodomain of about 200 amino acids (aa), whereas Gc, with an ectodomain of ca. 900 aa long, is about twice the size of its Gc counterparts from other bunyaviruses. While Gn and the C-terminal half of Gc are relatively well conserved across OBVs, the Gc N-terminal half is not (Fig. 1a). Importantly, deletions of the N-terminal half of Gc of Bunyamwera virus (BUNV, the OBV type species—which also gave the name to the whole *Bunyavirales* order) resulted in a mutant virus that was still able to replicate in cell culture[14], indicating that its fusion protein remained functional.

Despite the importance of OBVs, which constitute the largest genus within the *Bunyavirales* order with more than 170 named viruses in 18 serogroups distributed across the planet[15,16], little is known about the structural organization of the infectious OBV particle. Low-resolution electron cryo-tomography studies on BUNV revealed pleomorphic particles displaying prominent trimeric spikes[17]. Yet, the resolution of the reconstruction did not provide insight into the molecular details of the spike's architecture.

Here, we describe the X-ray structure of the N-terminal half of SBV Gc, showing that it is composed of two domains with a novel fold: an N-terminal α-helical head domain followed by a stalk composed of two tandem β-sheet subdomains. We also report the X-ray structures of the head domains of OROV, LACV, and BUNV Gc, which, together with the SBV head-and-stalk structures, show that these domains completely account for the prominent trimeric spikes of the OBV envelope. We further demonstrate that the recombinant head–stalk fragment of SBV

Gc is able to deplete the neutralizing activity of sera from reconvalescent ruminants—the natural SBV hosts. Moreover, we show that both active and passive immunization against this construct protects from disease symptoms in a mouse model. Finally, the X-ray structures of two neutralizing monoclonal antibodies against SBV in complex with the Gc head domain show that they target the trimer interface of the spike. Integrating these findings with available aa sequence data on the OBV envelope proteins, we show that the SBV glycoprotein spike specifically mutates at key structural and antigenic sites during persistent fetal infection of ruminants.

## Results

**The N-terminal half of OBV Gc is composed of two domains.** We crystallized the N-terminal half of SBV Gc (aa 465–874 in the glycoprotein precursor, Fig. 1a) produced in recombinant *Drosophila* S2 cells. The best crystals diffracted anisotropically to 2.5–2.0 Å resolution, which allowed the X-ray structure determination and refinement to a final $R_{free}$ value of 0.24 (Supplementary Table 1). The structure showed two rigid domains separated by a hinge, adopting a V-shaped conformation in the crystal (Fig. 1b). The N-terminal head domain (aa 465–702) constitutes the first arm of the V. It is an α-helical bundle with approximate dimensions of $6.5 \times 4.0 \times 3.5$ nm in size with a molecular mass of 27 kDa. It is internally stabilized by four disulfide bonds and carries two N-linked carbohydrate chains at positions N493 and N686, protruding from the same face of the ellipsoidal head domain. Of note, the glycan chain attached to N493, which was earlier shown to be essential for protein folding and secretion[18], is well ordered through stabilization by the local protein environment (Supplementary Fig. 1) and not by crystal packing. The rod-shaped second domain, which we term stalk (see below), has dimensions of $7.0 \times 3.0 \times 2.5$ nm and extends from the vertex of the V-shaped molecule. The stalk can be further divided into two subdomains (aa 703–798/ 799–874). Despite having low aa sequence identity (18%) with each other, the two stalk subdomains share a common fold with a root-mean-square deviation (RMSD) of 3.2 Å over 64 Cα atoms (Fig. 1c, d), indicating that they derive from gene duplication. Each stalk subdomain has a molecular mass of ~19 kDa and comprises two antiparallel β sheets packing against each other. Moreover, each stalk subdomain is internally stabilized by three conserved disulfide bonds, with the corresponding cysteine residues conserved between them and across OBVs (Fig. 1d). A search of the Protein Data Bank with the Dali protein structure comparison server[19] showed that head-and-stalk domains have novel folds, without significant structural similarity to any known structure.

Inspection of the available aa sequences shows that all OBVs encode two tandem stalk subdomains, suggesting that the gene duplication event that resulted in the stalk must have occurred early in the evolution of these viruses, or even before its incorporation into an OBV precursor via lateral gene transfer. Since then, a strong selective advantage appears to have favored the maintenance of both subdomains. It is possible that the presence of two stalk subdomains is required for proper positioning of the head domain in order to facilitate its interaction with host cell factors or with neighboring Gc molecules to stabilize the OBV particle. The latter is supported by additional structural data outlined below.

**The N-terminal domains of Gc form the trimeric OBV spike.** In addition to the variable region of SBV Gc, we determined the crystal structures of the Gc head domains from BUNV (aa 478–721), LACV (aa 477–722) and OROV (aa 482–702) to resolutions ranging between 2.1 Å and 2.9 Å and refined the

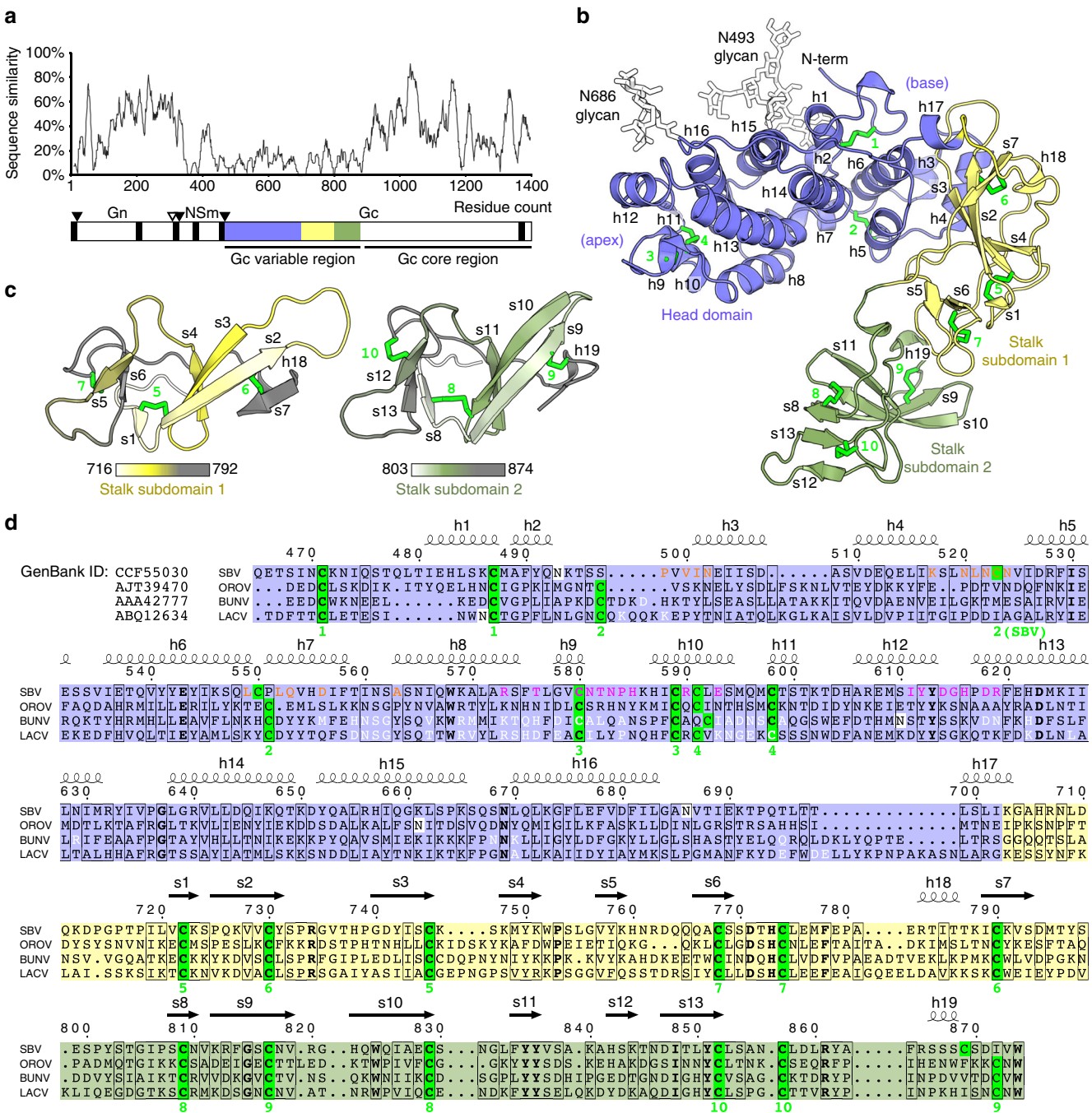

**Fig. 1** The N-terminal variable half of OBV Gc is composed of two domains. **a** Relative amino acid sequence similarity across 100 OBVs over the glycoprotein precursor. Putative transmembrane regions are indicated as black bars. Three secretion signal peptidase cleavage sites are indicated as filled triangles and an additional protease cleavage site is indicated as an empty triangle[52]. **b** Crystal structure of the variable region of SBV Gc. The head domain is shown in blue, and the two stalk subdomains are shown in yellow and green. The two carbohydrate chains were not originally part of the structure of this deglycosylated protein, and were added to this illustration based on the crystal structure of the SBV head domain in complex with scFv 1C11 (Fig. 5 and Supplementary Fig. 1). Disulfide bonds are shown in bright green and are numbered sequentially. Secondary-structure elements are labeled with h for helices and s for β strands. **c** The two stalk subdomains are aligned to each other for structural comparison. They are colored in a gradient from white at their N-terminus to gray at their C-terminus. **d** Sequence alignment of the variable region of Gc from SBV, OROV, BUNV, and LACV. Secondary-structure elements and sequence numbering of SBV are given above the alignment; disulfide bonds are numbered sequentially below the alignment. Strictly conserved residues are boldfaced, similar residues are boxed. Cysteines involved in disulfide bonds are highlighted on bright green background; N-glycosylation sites are highlighted on white background. Residues at the epitopes of mAbs 1C11 and 4B6 are indicated in magenta and orange font, respectively. Residues at the trimer interfaces of BUNV and LACV are indicated in white font. The sequences of the two stalk subdomains are aligned to each other in the last two rows of the alignment

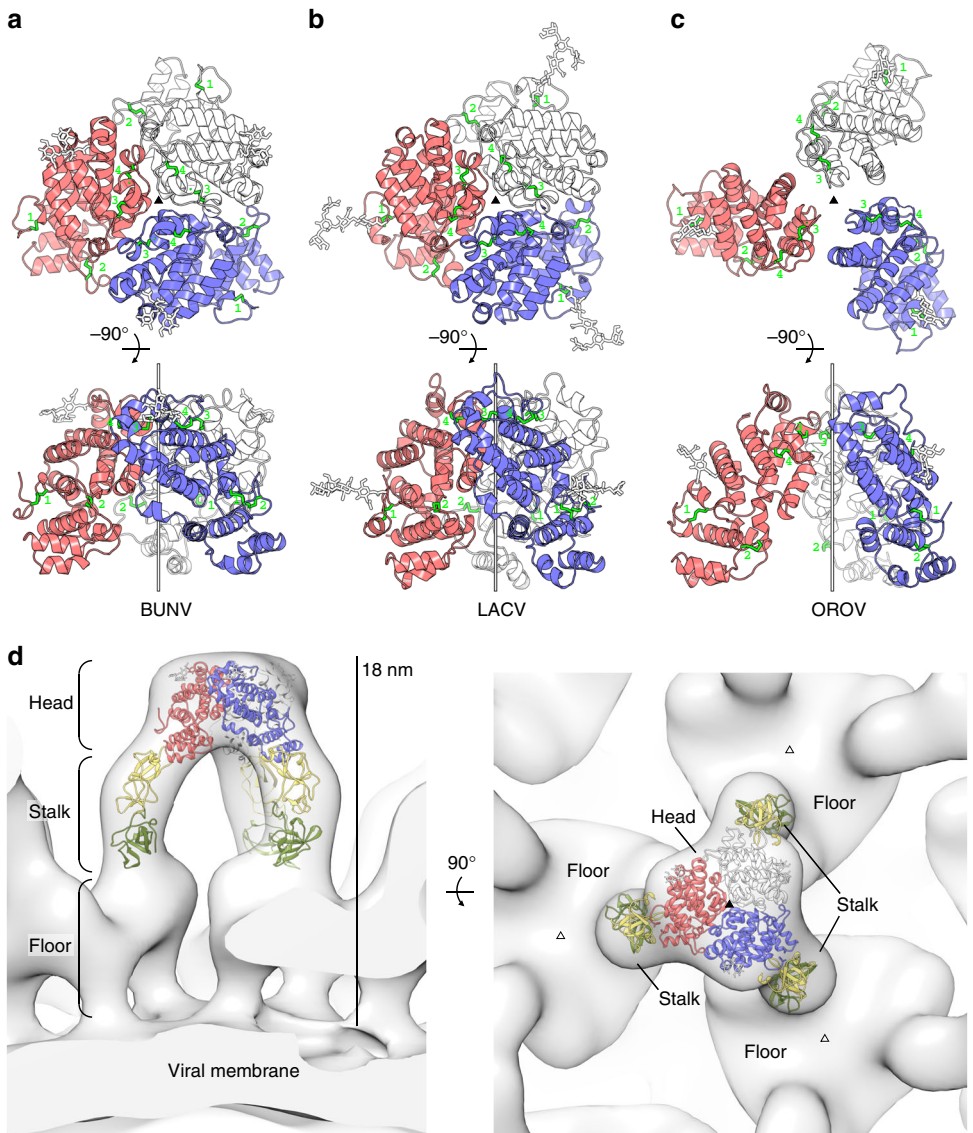

**Fig. 2** The head and stalk domains form the trigonal pyramid of the OBV envelope. **a–c** Trimeric head domain assemblies as found in the crystal structures for **a** BUNV, **b** LACV, and **c** OROV. Trimers are shown in top view (top) and in side view (below). The threefold crystallographic axes are indicated as filled triangles (top views) and as vertical bars (side views). The three identical protomers are shown in blue, red, and white, respectively. Disulfide bonds are shown in bright green; carbohydrate chains are shown as white sticks. **d** Position of head and stalk within the electron cryo-tomography map of the BUNV glycoprotein spike[17]. The displayed model is composed of the trimeric BUNV head domain structure as in panel a and of the stalk from SBV as in Fig. 1b. The spike is shown in side view (left) and in top view (right). The two non-equivalent threefold symmetry axes within the local structure of the envelope are indicated as filled and empty triangles (right)

resulting atomic models to $R_{free}$ values between 0.21 and 0.23 (Fig. 2a–c and Supplementary Table 2). The aa sequence identity among the head domains of all the four viruses is low and ranges from 29% (BUNV:LACV) to 17% (SBV:BUNV). Nevertheless, they share a common tertiary structure with RMSD values between 2.0 Å over 232 Cα atoms (BUNV:LACV) and 3.7 Å over 224 Cα atoms (LACV:SBV) (Supplementary Table 3). The head domains of BUNV, LACV, and OROV contain each a single asparagine-linked carbohydrate chain (Figs. 1d and 2a–c) and although the positions of the respective glycosylation sites along the aa sequences are not conserved, they project from the same face of the domain carrying the two glycan chains in SBV. It is thus very likely that this gly-cosylated face is exposed to the environment in the context of mature virus particles (Supplementary Fig. 3d).

The recombinant head domains of Gc from all four viruses behaved as monomers in solution, both at pH 8.0 and at pH 5.5.

The presence of 150 mM potassium, which was recently shown to promote BUNV infection in host endosomes[20], did not affect the oligomeric state (Supplementary Fig. 2a). Nevertheless, the Gc head domains of BUNV and LACV both formed trimeric assemblies involving the same trimerization contacts in the crystals (Fig. 2a, b). The buried surface area at the trimer interface amounts to 1422 Å² and 1430 Å² on each protomer for BUNV and LACV, respectively. Whereas the atomic details at these interfaces differ between the two viruses, they comprise mainly hydrophilic residues in the region between helices h7 and h11 (Fig. 1d and Supplementary Fig. 3a) at the side opposite to the glycosylated surface. The concurrent observation of two very similar trimeric assemblies in two independent crystal structures strongly suggests that the trimers are physiologically relevant.

The electron cryo-tomography structure of the complete BUNV envelope glycoprotein coat is available at a resolution of

3 nm[17]. The most prominent feature of this structure is a tripodal pyramid protruding 18 nm from the viral surface, with a floor region coating the viral membrane and connecting three adjacent pyramidal spikes. While the low resolution of this reconstruction does not allow for a sequence-correlated domain assignment per se, our crystal structure of the trimeric BUNV Gc head domain fits unambiguously into the most membrane-distal region of this pyramid (correlation coefficient = 0.92; Fig. 2d). A second threefold symmetry axis is also present in the floor, but the density in this region is incompatible with our crystal structure. Consequently, the three legs of the tripodal pyramid must each accommodate the stalk (correlation coefficient = 0.88; Fig. 2d). These conclusions imply that the conserved Gc core domains, together with three Gn ectodomains, are located in the floor, connecting three adjacent spikes in close proximity to the viral membrane.

The unambiguous fit of the head domain at the tip of the spike indicates that its orientation relative to the stalk is different between the OBV particle and the V-shaped crystal structure (compare Fig. 1b and Fig. 2d), suggesting that the head/stalk interface can act as a hinge to allow for a range of conformations. This notion is supported by the observation that the tripodal pyramid collapses upon exposure of BUNV particles to pH 5.1[17]. The dissociation of the trimers at low pH is likely triggered by protonation of residue H590 in BUNV and in LACV, which would lead to electrostatic repulsion with positively charged side chains on the neighboring protomer (Supplementary Fig. 3b–c). Although the positions of histidine residues at the interface are not strictly conserved, there is at least one histidine with presumably similar function present at each of the inferred trimerization sites of SBV and OROV (Fig. 1d).

The OROV Gc head domain did not crystallize as a closed trimer, but its individual protomers were arranged around a threefold crystallographic axis in a similar fashion as the trimer subunits of the BUNV and LACV head domains (Fig. 2c and Supplementary Fig. 4). This exception is probably not related to the pH, as all the three head domain structures were obtained in the same pH range between pH 7.5 and pH 8.5. The three protomers of the open OROV trimer are held apart from each other via crystal-packing restraints. Although this arrangement may have occurred coincidentally, it is conceivable that the molecules transiently formed closed trimers at an intermediate step of crystal growth, which eventually influenced their final packing pattern. Such a process would be consistent with our observation that trimerization is very weak in solution (Supplementary Fig. 2a). Yet, the viral envelope provides a very high local concentration of oriented subunits that may shift the equilibrium to favor the interaction. Of note, the observed trimer interface area is well conserved in BUNV, LACV, OROV, and SBV when each virus is compared with its closest orthologs, in contrast to the exposed surface of the trimer head, supporting the functional importance of trimerization across OBVs (Supplementary Fig. 3d). The low trimer affinity of the Gc head domain may facilitate efficient glycoprotein coat disassembly to allow for cell entry.

**SBV neutralizing antibodies target the spike in native hosts.** The jutting three-dimensional structure of the N-terminal half of Gc predestines it as an exquisite target for the humoral adaptive immune response of the host. Indeed, it was shown that the majority of envelope-specific murine mAbs raised against LACV or SBV infectious virus recognize the head domain[21,22]. In order to test whether this immune bias for the head domain also occurs during acute primary infection of a native mammalian host, we performed antibody-depletion experiments on sera from three

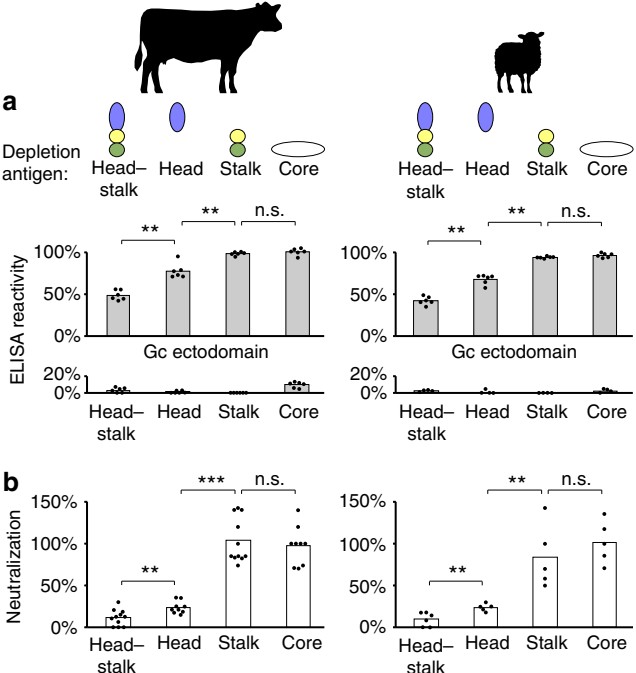

**Fig. 3** The SBV Gc head domain is the major target of neutralizing antibodies in its natural hosts. **a** Antibody subpopulations were depleted from three cattle sera (left) and one sheep serum (right) collected 28 days after experimental infection with SBV. The above-indicated recombinant depletion antigens were head–stalk (aa 465–874), head (aa 465–702), stalk (aa 694–874), and core (aa 890–1326). The histogram shows ELISA reactivity after depletion with the full Gc ectodomain (aa 465–1326, top) or with the respective depletion antigens (below). Bars represent mean percentages of mock-depleted sera, the optical density values of which ranged robustly between 3.2 and 3.6 for reactivity with the Gc ectodomain. Owing to saturation effects, ELISA reactivity is likely not linearly proportional to the absolute binding activity in a sample. **b** Virus neutralization after depletion with the indicated antigens for the three cattle sera (left) and the sheep serum (right). Bars represent mean percentages of mock-depleted sera. Dots represent technical replicates. Statistical analysis of the neutralization data was performed using the Mann–Whitney rank sum test. **$P < 0.01$; ***$P < 0.001$; n.s. not significant ($P > 0.05$). Source data are provided as a Source Data file

reconvalescent cattle and one sheep obtained 28 days after experimental infection with SBV.

We used four recombinant antigens to deplete domain-specific antibody subsets from the sera: the N-terminal head–stalk construct, the head domain only, the stalk only, and the conserved Gc core ectodomains. All four constructs behaved correctly and eluted as monomers in solution (Supplementary Fig. 2b), indicating that their native fold was preserved. After depletion with each of the four antigens, we determined the remaining ELISA reactivity of the sample with the entire Gc ectodomain (Fig. 3a, top). To confirm efficient depletion of each antibody subpopulation, we also determined the ELISA reactivity with the same domain used for depletion (Fig. 3a, bottom). The remaining neutralization activity after depletion was determined in a microneutralization assay using SBV isolate BH80/11–4 on BHK-21 cells (Fig. 3b).

The results of the experiments with cattle were virtually identical to those with the sheep. Depletion with the head–stalk construct resulted in a strong reduction in ELISA reactivity with the entire Gc ectodomain and almost completely abrogated the neutralization activity of the sera. Depletion with the head

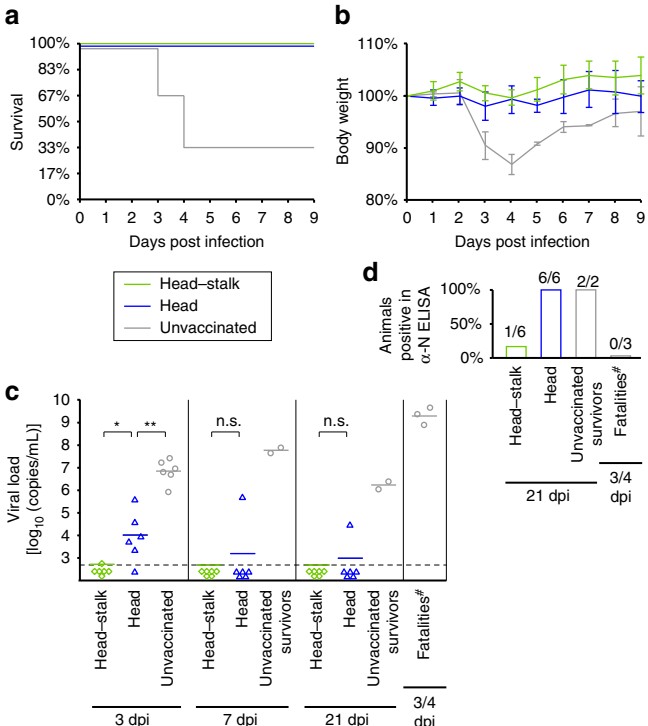

**Fig. 4** Immunization with the SBV head–stalk fragment protects IFNAR$^{-/-}$ mice from viremia. **a, b** Survival curves (**a**) and relative body weight development (**b**) of vaccinated and unvaccinated mice over the first 9 days post infection ($n = 6$ per group). The immunization antigens were head–stalk (aa 465–874) and head (aa 465–702). Weights are given as mean percentage ± s.d. **c** Viral genome loads at 3, 7, and 21 days post infection (dpi) or on the day when an unvaccinated animal had to be euthanized (fatalities). Genome equivalent concentrations were determined in EDTA–blood samples by S segment-based RT-qPCR. Horizontal bars represent mean values. The dashed line indicates the detection limit ($5 \times 10^2$ copies/mL). Statistical analysis was performed using the Mann–Whitney rank sum test. *$P < 0.05$; **$P < 0.01$; n.s. not significant ($P > 0.05$). **d** Percentage of animals positive for N protein-specific antibodies at 21 dpi or on the day when an unvaccinated animal had to be euthanized, as determined by ELISA. The numbers of seroconverted animals in each category is indicated. #One unvaccinated animal naturally succumbed to the infection and was thus not available for blood analysis. Source data are provided as a Source Data file

domain alone resulted in a slightly less pronounced effect on both, ELISA reactivity with the Gc ectodomain and neutralization activity. In contrast, when we used just the stalk domain for depletion, we detected neither a reduction in ELISA reactivity with the Gc ectodomain nor in neutralization activity. It is thus possible that conformational epitopes at the head/stalk junction, which are neither fully covered by the head nor by the stalk, render the head–stalk construct a more efficient depletion antigen than the head domain alone. Finally, depletion with the conserved C-terminal core ectodomains also resulted in no measurable reduction in ELISA reactivity with the entire Gc ectodomain or in neutralizing activity. These findings imply that the local antigenicity and immunogenicity within the glycoprotein spike increase with the distance from the viral membrane, which likely relates to local accessibility in the context of the virus particle. Thus, in agreement with the previous findings in mice immunized with infectious virus[21,22], we show that the well-exposed Gc head domain is also the major target of neutralizing antibodies in native host animals after acute primary infection with SBV.

**SBV Gc subunit vaccine trial in IFNAR$^{-/-}$ mice.** In order to evaluate the translatability of the above findings into potential clinical applications, we next performed a subunit vaccine trial with IFNAR$^{-/-}$ mice as a model for SBV infection. Earlier OBV subunit vaccine trials had demonstrated that a correctly folded fragment spanning the N-terminal 250 aa of SBV Gc (here shown to correspond to the head domain) conferred protection from clinical symptoms, whereas preparations using a misfolded counterpart did not[23]. Given the superior performance of the head–stalk construct with respect to the head domain only in our depletion experiments, here we set out to test the efficacy of this larger construct in protecting IFNAR$^{-/-}$ mice from lethal SBV infection.

We immunized groups of six mice twice with 20 µg of either the head–stalk construct or the head domain alone, supplemented with 20 µL of Emulsigen adjuvant. A control group of six mice was mock-immunized with PBS. Three weeks after the second immunization, we challenged the animals with $10^5$ TCID$_{50}$ SBV isolate BH619/12 and collected blood samples for viral RNA quantification on days 3, 7, and 21. We immediately euthanized mice showing severe clinical signs and killed the surviving mice on day 21 to test their sera for antibodies against the SBV nucleoprotein N. This highly immunogenic antigen is produced in large quantities during OBV replication, such that the presence of antibodies against N is a sensitive marker of viral replication within the host[24,25].

Four of six animals in the control group developed severe clinical signs in response to the challenge infection and either died or had to be euthanized on days 3 or 4 post infection (Fig. 4a, b). High viral genome loads at all sampling time points confirmed a strong infection of all control animals, including the two survivors (Fig. 4c). In support of this, both surviving animals had mounted an N protein antibody response as monitored by ELISA at the end of the experiment (Fig. 4d). In contrast to the control group, all vaccinated animals survived the challenge infection without significant weight loss during the critical first 9 days post infection (Fig. 4a, b). Thus, immunization with both immunogens provided protection from clinical symptoms in our experimental setting. Yet, differences in viral genome loads indicate that the head–stalk construct is a more effective subunit immunogen (Fig. 4c). Notably, this construct conferred virtually sterile immunity, with only one animal displaying a very low transient viremia close to the RNA detection limit on day 3. No viral RNA could be detected later. This was also the only animal to develop antibodies against the N protein in this group (Fig. 4d). In contrast, five of six animals immunized with the head domain were clearly positive for viral RNA on day 3. Moreover, all animals in this group had developed antibodies against the N protein by the end of the experiment, demonstrating that viral replication indeed took place in these animals upon challenge infection (Fig. 4d).

Our results are consistent with the earlier trial[23] in what concerns immunization with the head domain; however, we now demonstrate that the larger head–stalk construct has even higher efficacy, leading to no detectable viral replication upon SBV challenge in most immunized animals. While it is possible that potentially favorable pharmacokinetic properties of this construct contribute to its superior performance, the presentation of additional epitopes as compared with the head domain alone likely also induces a broader and thus more potently protective antibody response.

**Two antigenic sites on SBV Gc map to the trimer interface.** To better understand antigen recognition by neutralizing antibodies against SBV, we took advantage of a panel of 10 envelope-specific murine mAbs raised against SBV infectious virus[22,26].

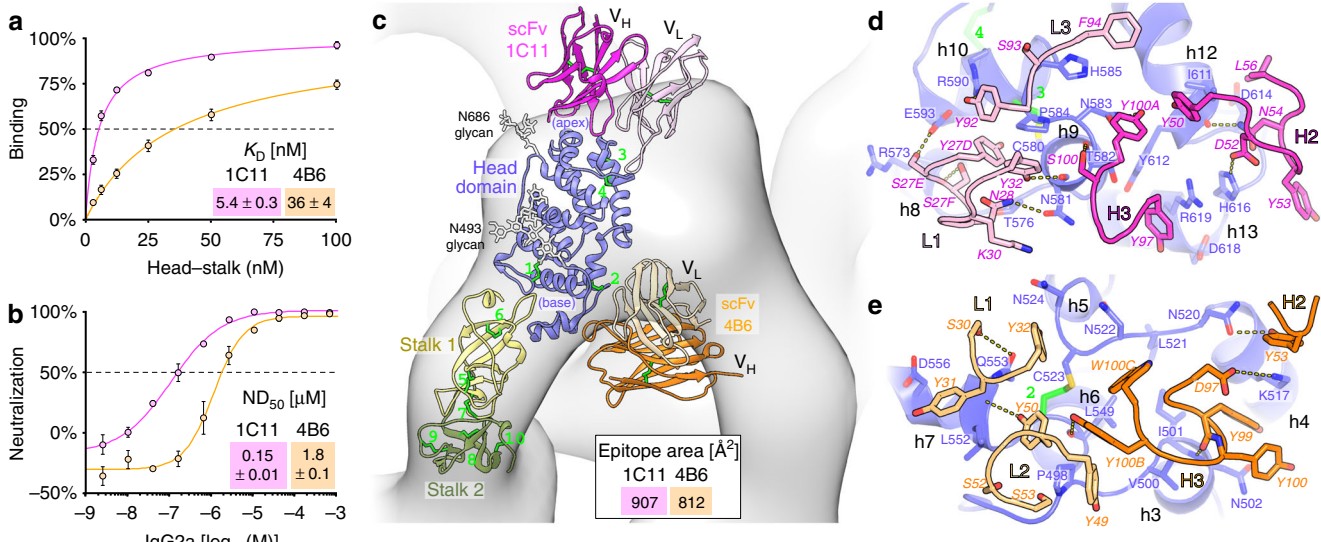

**Fig. 5** The two previously identified antigenic sites on SBV Gc map to the trimer interface. **a** Steady-state binding curves for the recombinant Gc head–stalk construct binding to immobilized mAbs 1C11 and 4B6 in biolayer interferometry. Data were acquired in duplicates and are displayed as mean ± s.d. Source data are provided as a Source Data file. **b** Focus reduction neutralization (FRNT) of ~750 focus-forming units per mL SBV isolate BH80/11-4 with mAbs 1C11 and 4B6 on Vero cells. Data were acquired in duplicates and are displayed as means ± s.d. Negative neutralization values indicate a slight enhancement of infection with respect to a control without antibodies. Source data are provided as a Source Data file. **c** Model of the single-chain variable fragments (scFvs) of 1C11 (magenta) and of 4B6 (orange) bound to a single protomer of the SBV spike that is docked into the electron cryo-tomography map of the trimeric BUNV glycoprotein spike[17]. The illustration was compiled from two separate crystal structures of the head domain in complex with each scFv. **d** Detail of the interaction interface with 1C11. **e** Detail of the interaction interface with 4B6. Residues on both sides of the interface are labeled and are shown as sticks with oxygen atoms in red and nitrogen atoms in blue. Secondary-structure elements and disulfide bonds are labeled as in Fig. 1

Preliminary epitope mapping with these mAbs had identified two major antigenic sites[18] on the Gc region corresponding to the head domain. For further characterization, we selected two mAbs from this panel, 1C11 and 4B6—both IgG2a isotypes—one for each of the two antigenic sites.

The recombinant SBV head–stalk construct bound to these mAbs with dissociation constants of $K_D(1C11) = 5.4 ± 0.3$ nM and $K_D(4B6) = 36 ± 4$ nM (mean ± s.d., Fig. 5a). Furthermore, the two mAbs neutralized SBV on Vero cells in a focus reduction neutralization (FRNT) assay with $ND_{50}(1C11) = 0.15 ± 0.01$ μM and $ND_{50}(4B6) = 1.8 ± 0.1$ μM (mean ± s.d., Fig. 5b). These values fall into the same range as previously reported OBV plaque reduction neutralization (PRNT) titers of a set of 18 envelope-specific mAbs raised against LACV (median $ND_{50} = 10.4$ μM, lowest $ND_{50} = 0.16$ μM[21]). Sequencing of the variable regions of 1C11 and 4B6 revealed that their aa sequences are close to their respective murine germline, with only 4 and 3 aa substitutions in the V-regions, respectively (Supplementary Fig. 5).

We co-crystallized single-chain variable fragments (scFvs) of 1C11 and 4B6 in complex with the SBV Gc head domain. The crystals diffracted anisotropically to 2.8–3.6 Å resolution for 1C11 and to 3.2–4.0 Å resolution for 4B6. We determined the structures by molecular replacement and refined the atomic models to final $R_{free}$ values of 0.26 and 0.23, respectively (Fig. 5c and Supplementary Table 1). The structures showed that both mAbs recognize conformational epitopes on a single antigen monomer. 1C11 directly binds to the apical end of the head domain near the N686 glycosylation site. Its epitope has an area of 907 Å² and is centered on the third disulfide bond, with contributions from the regions around helices h8–h10 and h12–h13 (Figs. 1d and 5d). 4B6, on the other hand, binds to the opposite, basal end of the head domain, near its connection to the stalk. Its epitope covers 812 Å² of surface area and is centered on the second disulfide bond, with contributions from the region around helices h3–h7 (Figs. 1d and 5e).

The structures further revealed that the epitope of both antibodies overlaps with the Gc head trimer interface and is not accessible in the context of the trimeric spike (Fig. 5c). The fact that both antibodies neutralize virus particles therefore suggests that the SBV spikes are dynamic and transiently expose this epitope. Indeed, the affinity of the antibodies for the head monomer (Fig. 5a) is much higher than the trimer affinity of the head domain, as we could not detect a head trimer by size exclusion chromatography. Trimerization appears to require a high local concentration of Gc, as is the case at the virion surface. In addition, since OBV particles are pleomorphic and their triangular glycoprotein lattice exhibits multiple imperfections to follow the particle's curvature[17], it is possible that permanently monomeric Gc protomers are also present at the virion surface. The neutralization mechanism of our antibodies is therefore likely based on steric hindrance of host cell attachment or of membrane fusion, while a potential destabilization of the envelope glycoprotein lattice may also contribute to the neutralization effect.

## 1C11 and 4B6 protect IFNAR$^{-/-}$ mice from lethal SBV infection.

In order to test if our two mAbs 1C11 and 4B6 were also protective in vivo, we again applied our IFNAR$^{-/-}$ mouse model for SBV infection. Groups of six mice each were treated with 200 μg of either mAb 1C11, 4B6, or an equal volume of PBS, 24 h ahead of challenge infection with $10^5$ TCID$_{50}$ of SBV isolate BH619/12. The subsequent analysis was performed as for the subunit vaccine trial described above. While all six animals in the mock-treated control group developed severe clinical signs and either died or had to be euthanized 3 or 4 days post infection, all animals that had received a head-specific mAb survived the challenge infection without clinical signs (Fig. 6a, b). Although five of six animals in the 1C11 group and all animals in the 4B6 group became viremic by day 3, and also displayed infection-induced seroconversion against the N protein on day 21, their

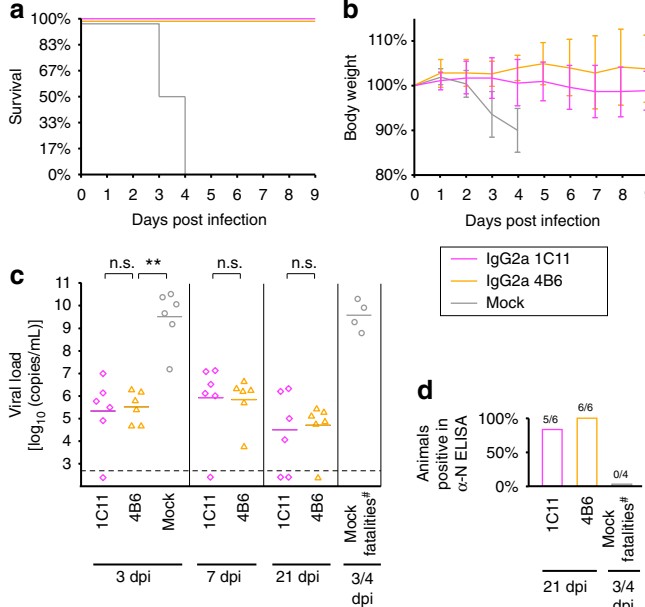

**Fig. 6** 1C11 and 4B6 protect IFNAR$^{-/-}$ mice from lethal SBV infection. **a**, **b** Survival curves (**a**) and relative body weight development (**b**) of antibody-treated and untreated mice over the first nine days post infection ($n = 6$ per group). The mAbs had been administered 24 h prior to challenge infection. Weights are given as mean percentage ± s.d. **c** Viral genome loads at 3, 7, and 21 dpi or on the day when an unvaccinated animal had to be euthanized (fatalities). Horizontal bars represent mean values. The dashed line indicates the detection limit. Statistical analysis was performed using the Mann–Whitney rank sum test. **P < 0.01; n.s. not significant ($P > 0.05$). **d** Percentage of animals with N protein-specific antibodies at 21 dpi or on the day when an unvaccinated animal had to be euthanized. The numbers of seroconverted animals in each category is indicated. #Two of six mock-treated animals naturally succumbed to the infection and were thus not available for blood analysis. Source data are provided as a Source Data file

viral genome loads stayed significantly below those in the unprotected group (Fig. 6c, d).

These results demonstrate that both mAbs can protect IFNAR$^{-/-}$ mice from lethal SBV infection in vivo. In addition to direct neutralization, this protection may be reinforced by complement- and effector cell-dependent mechanisms, although a detailed examination of such antibody effector functions is beyond the scope of our current study. Our results suggest that antibodies with properties similar to 1C11 and 4B6 also contribute to virus clearance in the course of OBV infections in the natural hosts. Similar antibodies may thus also exert a selection pressure for OBV evolution in the field, as can be inferred from the natural aa sequence drift of SBV in persistently infected ruminant fetuses, as analyzed below.

**Antigenic variation in naturally infected ruminant fetuses.** Whereas SBV infection is rapidly cleared from the blood of immunocompetent adult animals, the virus can persistently replicate to high titers in the unborn. This causes abortion, stillbirth, or birth with severe congenital abnormalities in the newborn, which typically include arthrogryposis and hydranencephaly[27]. Virus from prenatally infected lambs and calves was shown to accumulate mutations specifically in the region encoding the Gc head domain[28–30]. Our structural data now allow us to assess the SBV aa sequence variability in a site-specific manner and to derive initial functional interpretations.

SBV belongs to a group within OBVs that includes Sathuperi virus (SATV), Douglas virus (DOUV), Shuni virus (SHUV), and Aino virus (AINOV), which share an atypical second disulfide bond in their Gc head domain as opposed to the other known OBVs (Supplementary Data 1). Comparison of wild-type circulating isolates of these five viruses shows that the most conserved area on the surface of the Gc head domain is the putative trimer interface (Fig. 7a). Moreover, the stalk carries a high proportion of conserved residues at its contact area with the head domain. Both these findings are expected as protein interaction sites are generally better preserved in evolution than strictly solvent-exposed sites.

SBV from adult ruminants undergoing acute infections throughout Europe during the initial outbreak in 2011/2012 displayed an overall very high degree of aa sequence conservation (Fig. 7b and Supplementary Data 2). In contrast, SBV variants sequenced during the same time period from fetal or malformed newborn tissue displayed higher variability[27]. Mapping the aa changes observed in the latter variants to the structure of the head–stalk region of Gc revealed a strikingly non-random distribution (Fig. 7c and Supplementary Data 2). Intriguingly, the observed mutations tend to accumulate on helix h8 at the putative trimer interface and at the head–stalk interface. This process likely reflects an adaptation to the cellular and biochemical environment in utero. Of note, some of the most frequent mutations in SBV from aborted fetuses or malformed newborns are also located directly at the epitopes of our two mAbs 1C11 and 4B6 and at other positions that have not yet been assigned a function (Fig. 7c and Supplementary Data 2). Contrary to humans, ruminant fetuses develop an antibody response, and the observed mutations could potentially be attributed to immune escape, as SBV antibodies are frequently detected in malformed lambs and calves[31–33].

Irrespective of the exact nature of the selection pressures that drive this aa sequence drift, we suggest that many of the fetus-specific mutations are mainly of degenerate nature and do not reflect a gain of function. This notion is supported by occasional observations of virtually complete deletions of the Gc head domain in malformed lambs[34]. Similar spontaneous deletions were also observed in other OBVs, such as cell culture-passaged Maguari virus[35] and in mouse brain-passaged Koongol virus[36]. Despite this phenomenon, the high genetic stability of SBV in acutely infected adult animals suggests that the fetus-specific mutations are incompatible with completion of a full viral cycle in its insect and mammalian hosts. Whether or not the fetus-specific point mutations can revert in the insect vector upon uptake from prenatally infected newborns is not yet clear. It also remains to be investigated if similar host-specific adaptation processes occur in the viral cycle of OBVs other than SBV.

## Discussion

The N-terminal variable half of the Gc glycoprotein is a unique feature of OBVs among bunyaviruses. Our structural analysis has revealed that this region is composed of two domains: a 27 -kDa α-helical head domain and a stalk that is divided into two identically folded 19 -kDa tandem β-sandwiches. Reverse genetics studies on BUNV have shown that deletions of either the head domain alone, the head domain together with the first stalk subdomain or the whole N-terminal variable half of Gc can be tolerated in vitro[14]. On the other hand, partial N-terminal deletions of the head domain or deletions of the complete head domain together with partial deletions of the first stalk subdomain were not viable. These data are in good agreement with our structure-based modular domain annotation.

The evolutionary origin of the N-terminal unique domains of OBV Gc remains obscure, as they display novel folds with no

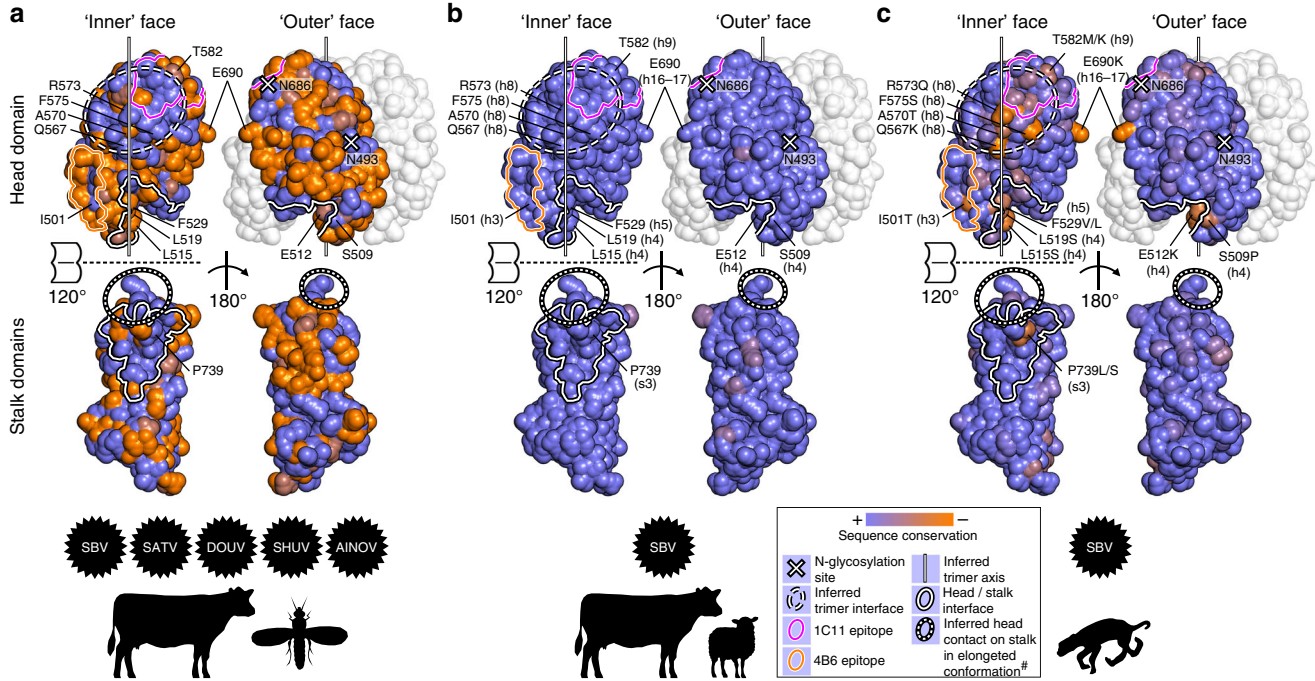

**Fig. 7** The N-terminal variable half of SBV Gc mutates at structurally and antigenically critical sites in ruminant fetuses. **a** Relative surface conservation among the SBV group of OBVs, including SATV, DOUV, SHUV, and AINOV (see main text), mapped to the structure of the N-terminal half of SBV Gc. The sequences represent viruses from adult viremic cattle or from midges. **b** Relative surface conservation among 13 unique SBV aa sequences from adult viremic ruminants collected in 2011 or 2012. **c** Relative surface conservation among 38 unique SBV aa sequences from malformed newborn ruminants collected in 2011 or 2012 (Supplementary Data 2). The relative surface conservation is color-coded with blue for high similarity and orange for high divergence. Residues with high mutation frequency in aborted fetuses or malformed newborns are labeled and their nearest secondary-structure elements in the amino acid sequence are given in brackets. In all panels, the head domain is lifted off the stalk in order to expose the head/stalk interface area. #In the elongated conformation of the trimeric spike the footprint of the head on the stalk is shifted (dotted line) with respect to the V-shaped conformation of the crystal structure (closed line). The footprint of the stalk on the head is similar between the two conformations

similar structures in the Protein Data Bank. The head domain forms low-affinity trimers on the viral envelope that are recapitulated in the crystal contacts for BUNV and LACV. Yet, the observations that the variable region is dispensable for virus growth in vitro and that it specifically accumulates mutations during high-titer replication in mammalian fetal tissue in vivo suggest that trimerization may play a conditionally nonessential structural role in the stabilization and assembly of the viral envelope. Furthermore, the head domain is well exposed to the environment and is a major target of the humoral adaptive immune system of the OBV mammalian hosts, indicating that the trigonal pyramid of the spike shields the functionally more critical membrane fusion core machinery from the host's antibody response. The fusion machinery underlying the OBV spike would thus consist of the indispensable C-terminal half of Gc together with Gn, which we propose to reside in the floor connecting three adjacent spikes in the OBV particle (Fig. 2d). We hypothesize that shielding of the fusion machinery from the humoral adaptive immune system allows its evolutionary optimization for function by limiting antigenic drift, whereas the burden of escaping the immune pressure lies primarily on the projecting variable domains.

With regard to potential applications in human or veterinary medicine, the N-terminal variable half of Gc is a promising lead construct for OBV subunit vaccine design. Our immunization of IFNAR$^{-/-}$ mice with the SBV head–stalk construct efficiently inhibited viremia, whereas the head domain alone was less potent in this respect. Moreover, monoclonal antibodies against the Gc head domain could become efficacious therapeutic agents for the treatment of acute OBV infection. Given the very similar structural organization of different OBVs, our results therefore suggest a relatively straightforward approach to combat any emerging or re-emerging OBV in the future, both by immunotherapy and by vaccination with the head–stalk region of Gc.

## Methods

**Ethics statement**. All animal experiments were performed in accordance with the relevant guidelines and regulations (refs. LALLF M-VTSD/7221.3-1.1-004/12 and LALLF M-VTSD/7221.3-1.1-067/17 for the subunit immunogen trial and for the antibody protection trial, respectively). The experimental protocols were reviewed by the responsible state ethics commission and were approved by the competent authority (State Office for Agriculture, Food Safety and Fisheries of Mecklenburg-Vorpommern, Rostock, Germany).

**Cell lines**. BHK-21 (catalog number: CCLV 0164) and Vero 76 cells (catalog number: CCLV 0228) were obtained from the Collection of Cell Lines in Veterinary Medicine, Biobank of the Friedrich-Loeffler-Institut, Insel Riems, Germany. *Drosophila* S2 cells (catalog number: R69007) and FreeStyle 293-F cells (catalog number: R79007) were purchased from Thermo Fisher Scientific.

**Viruses**. SBV isolate BH80/11-4 was used for the microneutralization assay and for the focus reduction neutralization assay. SBV isolate BH619/12 was used for the challenge infection of mice. Virus stocks were prepared in BHK-21 cells in the minimal essential medium (MEM) with 5% FCS. Virus was harvested 72 h post infection with one cycle of freezing and thawing followed by centrifugation. Aliquots were stored at −80 °C until use.

**Subunit immunogen trial**. Twenty-one IFNAR$^{-/-}$ mice of C57BL/6 genetic background (B6.129S2-Ifnar1tm1Agt/Mmjax) were obtained from the specific pathogen-free breeding unit of the Friedrich-Loeffler-Institut. The animals were between 5 weeks and 10 months old and were randomly assigned to three groups of six animals. The remaining three animals were kept as environmental controls. Male, female, and young animals were distributed equally over all groups.

The mice were subcutaneously immunized twice with a delay of 21 days between the two immunizations. For each immunization, 20 μg of the respective recombinant immunogen was diluted in PBS and supplemented with 5% (v/v) Emulsigen adjuvans (MVP Technologies) in a final volume of 100 μL/mouse. Control animals were mock-vaccinated with an equal volume of PBS. Twenty-one days after the second immunization, the animals were subcutaneously infected with SBV isolate BH619/12 ($10^5$ TCID50/mouse). Following the challenge infection, the mice were weighed daily for 9 consecutive days and assessed for clinical signs. On days 3 and 7 post infection, EDTA–blood samples were collected for RT-qPCR analysis. Animals showing severe clinical signs were euthanized immediately on days 3 or 4 post infection. All remaining animals were killed 21 days post infection, and EDTA–blood as well as serum samples were collected for analysis.

**Antibody protection trial.** Twenty IFNAR$^{-/-}$ mice of C57BL/6 genetic background (B6.129S2-Ifnar1tm1Agt/Mmjax) were obtained from the specific pathogen-free breeding unit of the Friedrich-Loeffler-Institut. The animals were between 2 and 10 months old and were randomly assigned to three groups of six animals. The remaining two animals were kept as environmental controls. Male, female, and young animals were distributed equally over all groups.

The mAbs were administered at a concentration of 2 mg/mL in PBS in a volume of 100 μL/mouse by intraperitoneal injection, 24 h ahead of the challenge infection. Control animals were mock-vaccinated with an equal volume of PBS. The animals were subcutaneously infected with SBV isolate BH619/12 ($10^5$ TCID50/mouse). The subsequent experimental procedure was identical to the one described for the subunit immunogen trial.

**Recombinant protein preparation.** Except for the full-length murine IgG2a constructs (see below), all recombinant proteins were produced in *Drosophila* S2 cells (Gibco). All viral protein fragments were expressed from codon-optimized synthetic genes (Invitrogen) that were genetically fused to the N-terminal or C-terminal Strep tags. The N-terminal Strep tags were chosen for the variable regions of SBV Gc (aa 465–874) and LACV Gc (aa 477–911). Their sequences were EWSHPQFEKGG and EWSHPQFEKGGG, respectively. The C-terminal tags with the sequence GGWSHPQFEK were used for the SBV Gc head domain (aa 465–702), the BUNV Gc head domain (aa 478–721), the OROV Gc head domain (aa 482–702), and the two scFvs of 1C11 and 4B6. The two scFvs were constructed by joining the original coding sequences of the $V_H$ and $V_L$ regions by a (GGGGS)$_4$ linker. The antigens intended for serum depletion experiments contained a long spacer of 48 aa upstream of their C-terminal double Strep tags. This spacer was introduced to allow for efficient binding of the antigen/antibody complexes to Strep-Tactin beads without potential sterical restrictions from the bound antibodies. The complete sequence of the tag was: GGSQS DSRGG NGNGG GAGGN GGGSA AHIVM VDAYK PTKLE ENLYF QSAWS HPQFE KGGSGG GSGGS AWSHPQ FEK. All genes were cloned into the pMT expression vector (Invitrogen) with an N-terminal BiP secretion signal. Cloning primer sequences are provided in Supplementary Table 4.

Adherent cultures of *Drosophila* S2 cells (Gibco) were grown in Insect-XPRESS protein-free medium with L-glutamine (Lonza) supplemented with 25 U/mL penicillin/streptomycin (Gibco) at 28 °C. Expression plasmids were co-transfected with the selection plasmid pCoPURO[37] at a mass ratio of 20:1 using the Effectene transfection reagent (Qiagen) according to the manufacturer's instructions. Polyclonal stable S2 cell lines were established by selection with 7.5 μg/mL puromycin (Invivogen), which was added to the medium starting at 40 h after transfection. Cultures were expanded to 1 L of $10^7$ cells/mL in shaking flasks at 100 rpm and at 28 °C. Recombinant protein expression was then induced with 5 μM CdCl$_2$. Cell supernatants were harvested 1 week post induction, were concentrated to 50 mL on a 100 -kDa MWCO PES membrane (Sartorius), were pH-adjusted with 0.1 M Tris-Cl pH 8.0, were cleared from biotin with 15 μg/mL avidin, were cleared from precipitate by centrifugation at 4000×g for 15 min at 8 °C, and were used for affinity purification on a 5-mL Strep-Tactin Superflow hc column (iba Life Science). Proteins for vaccination and crystallization were further purified by gel permeation chromatography on a HiLoad Superdex 200 pg column (GE Healthcare) in 20 mM Tris-Cl pH 8.0, 150 mM NaCl. Protein concentrations were adjusted in 100 -kDa MWCO PES Vivaspin centrifugal concentrators (Sartorius).

**Crystallization of the SBV Gc variable region.** The N-terminal variable half of SBV Gc (aa 465–874) was enzymatically deglycosylated prior to crystallization. For this purpose, the target protein concentration was adjusted to 1 mg/mL and recombinant endoglycosidase H and endoglycosydase D were added at final concentrations of 10 μg/mL each. After incubation for 18 h at 24 °C, the deglycosylated target protein was separated from the His-tagged enzymes on a Strep-Tactin Superflow hc column (GE Healthcare). A final gel permeation chromatography step was performed in 20 mM Tris-Cl pH 8.0, 150 mM NaCl on a HiLoad Superdex 200 pg column (GE Healthcare). Optimal crystals were obtained by the hanging-drop vapor diffusion method: 0.5 μL of 12.3 mg/mL deglycosylated SBV variable region in 17.6 mM Tris-Cl pH 8.0, 132 mM NaCl, 0.5 M ammonium sulfate were added to 0.75 μL of reservoir solution containing 0.1 M MES pH 6.5, 5%w/v PEG 400, 2 M ammonium sulfate. The drops were equilibrated against reservoir solution

on siliconized glass slides (Hampton) for 2 days at 18 °C. Crystals were cryo-protected in a 50:50 v/v mixture of paraffin oil and silicon oil (Hampton) prior to conservation in liquid nitrogen. For preparation of samarium derivate samples for SAD phasing, native crystals were incubated in a drop containing 80 mM MES pH 6.5, 4% w/v PEG 400, 2 M ammonium sulfate, and 20 mM SmCl$_3$ for 2 h before cryo-cooling in a 50:50 v/v mixture of paraffin oil and silicon oil (Hampton) without back-soaking.

**Crystallization of the SBV Gc head domain with scFv 1C11.** The complex of the SBV Gc head domain (aa 465–702) with scFv 1C11 was purified from an equimolar mixture by gel permeation chromatography in 20 mM Tris-Cl pH 8.0, 150 mM NaCl on a HiLoad Superdex 200 pg column (GE Healthcare). Optimal crystals were obtained by the hanging-drop vapor diffusion method: 1.0 μL of a 21.7 mg/mL solution of the complex in 20 mM Tris-Cl pH 8.0, 150 mM NaCl was added to 0.75 μL of reservoir solution containing 0.1 M sodium acetate, 8% v/v 2-propanol, 25% w/v PEG 4 K. The drops were equilibrated against reservoir solution on siliconized glass slides (Hampton) for 5 days at 18 °C. Crystals were cryo-protected by immersion in 25% v/v glycerol, 75 mM sodium acetate, 6% v/v 2-propanol, 19% w/v PEG 4 K for 10 s prior to conservation in liquid nitrogen.

**Crystallization of the SBV Gc head domain with scFv 4B6.** The complex of the SBV Gc head domain (aa 465–702) with scFv 4B6 was purified from an equimolar mixture by gel permeation chromatography in 20 mM Tris-Cl pH 8.0, 150 mM NaCl on a HiLoad Superdex 200 pg column (GE Healthcare). Optimal crystals were obtained by the hanging-drop vapor diffusion method: 0.75 μL of a 22.8 mg/mL solution of the complex in 20 mM Tris-Cl pH 8.0, 150 mM NaCl were added to 0.75 μL of reservoir solution containing 0.1 M sodium acetate pH 4.6, 3.5 M sodium formate. The drops were equilibrated against reservoir solution on siliconized glass slides (Hampton) for 5 days at 18 °C. Crystals were cryo-protected by immersion in 143 mM sodium acetate pH 4.6, 5 M sodium formate for less than 30 s prior to conservation in liquid nitrogen.

**Crystallization of the BUNV Gc head domain.** Crystals of the BUNV Gc head domain (aa 478–721) were obtained by the sitting-drop vapor diffusion method: 0.2 μL of 22.2 mg/mL BUNV Gc head domain in 20 mM Tris-Cl pH 8.0, 150 mM NaCl were added to 0.2 μL of reservoir solution containing 0.2 M sodium acetate, 0.1 M Tris-Cl pH 8.5, 30% w/v PEG 4 K. Crystals appeared after 3 weeks of incubation at 18 °C and continued to grow for an additional 2 weeks. Crystals were cryo-protected by immersion for 10 s in 25% v/v glycerol, 0.15 M sodium acetate, 75 mM Tris-Cl pH 8.5, 22.5% w/v PEG 4 K prior to conservation in liquid nitrogen.

**Crystallization of the LACV Gc head domain.** Initial attempts to produce two different constructs of the LACV Gc head domain were unsuccessful. However, the complete variable region (aa 477–911) could readily be produced in S2 cells. Following affinity purification, the protein was further purified by gel permeation chromatography in 20 mM Tris-Cl pH 8.0, 150 mM NaCl on a HiLoad Superdex 200 pg column (GE Healthcare). As no crystals could be obtained with this construct, a sample of 0.7 mg/mL was proteolytically trimmed with 2 μg/mL trypsin (Sigma-Aldrich) for 30 min at 24 °C. The reaction was stopped with 5 μg/mL soybean trypsin inhibitor (Sigma-Aldrich), and the largest cleavage product with an apparent molecular mass of ~30 kDa was again purified by gel permeation chromatography in 20 mM Tris-Cl pH 8.0, 150 mM NaCl. Optimal crystals of this fragment were obtained by the hanging-drop vapor diffusion method: 0.75 μL of a 11.1 mg/mL protein sample in 20 mM Tris-Cl pH 8.0, 150 mM NaCl were added to 0.75 μL of reservoir solution containing 0.1 M HEPES pH 7.5, 10% v/v 2-propanol, 20% w/v PEG 4 K. The drops were equilibrated against reservoir solution on siliconized glass slides (Hampton) for 5 days at 18 °C. Crystals were cryo-protected by immersion in 17% v/v glycerol, 83 mM HEPES pH 7.5, 8.3% v/v 2-propanol, 16.6% w/v PEG 4 K for less than 30 s prior to conservation in liquid nitrogen.

**Crystallization of the OROV Gc head domain.** Optimal crystals of the OROV Gc head domain (aa 482–702) were obtained by the hanging-drop vapor diffusion method: 1.0 μL of 20.0 mg/mL OROV Gc head domain in 20 mM Tris-Cl pH 8.0, 150 mM NaCl was added to 1.0 μL of reservoir solution containing 0.5 M lithium sulfate, 15% w/v PEG 8 K. The drops were equilibrated against reservoir solution on siliconized glass slides (Hampton) for 2 days at 4 °C. Crystals were cryo-protected in a 50:50 v/v mixture of paraffin oil and silicon oil (Hampton) prior to conservation in liquid nitrogen. For preparation of samarium derivative samples for SAD phasing, native crystals were incubated in a drop containing 0.47 M lithium sulfate, 14% w/v PEG 8 K and 30 mM samarium(III) acetate for 16 h before cryo-cooling without back-soaking.

**X-Ray data collection and structure determination.** Native high-resolution X-ray diffraction data were recorded on synchrotron beamline PX1 at SOLEIL in St Aubin, France, with a PILATUS 6 M detector for the SBV Gc variable region, on beamline PX2 at SOLEIL with an EIGER X 9 M detector for both scFv complexes of the SBV Gc head domain as well as for the OROV head domain, on beamline ID30B at the ESRF in Grenoble, France, with a PILATUS 6 M detector for the

BUNV Gc head domain, and on beamline ID29 at the ESRF with a PILATUS 6 M detector for the LACV Gc head domain (Supplementary Tables 1 and 2). All datasets were processed with XDS[38] and Aimless[39]. Data corresponding to BUNV, LACV, and OROV proteins were truncated isotropically, whereas data corresponding to SBV proteins were truncated anisotropically using the STARANISO web server (Global Phasing Ltd.).

The structure of the OROV Gc head domain was determined by SAD phasing. A samarium derivate dataset of 180° (oscillation angle: 1°) was collected on our in-house Micromax 007 × -ray generator (Rigaku) with a standard rotating anode (wavelength 1.54 Å) in combination with a Mar345dtb image plate detector (Maresearch). The derivative dataset was processed with XDS[38] and Aimless[39]. Significant anomalous signal reached to a resolution of 2.9 Å, and SAD phasing was carried out with SHELXE[40]. The initial density map was readily interpretable and allowed for completion of the refinement through iterative manual model building in Coot[41] and automatic refinement in Phenix.Refine[42]. Phases were extended to the high-resolution native dataset in Phenix.MR. All further structures of Gc head domains were solved by molecular replacement with Phenix.MR[42]. Molecular replacement models for the scFv fragments were generated using the Phyre[2] web server for 3D homology modeling[43].

The structure of the complete variable region of SBV Gc was determined by SAD phasing in combination with molecular replacement. A samarium derivative dataset of 180° (oscillation angle: 0.1°) was collected at beamline PX1 of SOLEIL in St Aubin, France at the experimentally determined peak wavelength of 1.844 Å. The derivative dataset was processed with XDS[38] and Aimless[39]. Significant anomalous signal reached to a resolution of 3.3 Å and SAD phasing in combination with molecular replacement using the head domain from the scFv 1C11 complex structure was carried out with Phenix.Autosol[42]. The initial density map was readily interpretable and allowed for completion of the refinement through iterative manual model building in Coot and automatic refinement in Phenix.Refine[42]. Phases were extended to the high-resolution native dataset in Phenix.MR[42]. Stereo images of portions of the electron density maps are provided in Supplementary Fig. 6.

**Fitting of crystal structures into the Cryo-ET map**. The electron cryo-tomography map of the BUNV spike is available in the EMDataBank (entry: EMD-2352)[17]. The crystallographic trimer of our BUNV Gc head domain crystal structure and the SBV Gc stalk structure were sequentially fitted into the map using UCSF Chimera[44]. Correlation coefficients were calculated between simulated maps at 30 Å resolution for the crystal structures and the cryo-tomography map.

**Multi-angle static light scattering**. Purified recombinant proteins were subjected to size exclusion chromatography on a Superdex 200 10/300 column (GE Healthcare) equilibrated in the indicated buffers. Samples were pre-incubated in the respective running buffer for at least 1 h and were subsequently applied to the column at concentrations of 1 mg/mL. Separations were performed at 20 °C with a flow rate of 0.5 mL/min. Online multi-angle static light scattering analysis was performed with a DAWN-HELEOS II detector (Wyatt Technology). Online differential refractive index measurements were performed with an Optilab T-rEX detector (Wyatt Technology). Data were analyzed using the ASTRA software (Wyatt Technology).

**Ruminant serum depletion**. Depletion of antibody subpopulations was performed using the following recombinant Gc constructs: Gc head–stalk (aa 465–874), Gc head (aa 465–702), Gc stalk (aa 717–874), and Gc core (aa 890–1326). An unrelated protein was used as a mock control. The proteins carried a C-terminal 48-residues linker sequence and a double Strep tag as described above. The spacer was introduced to allow for efficient binding of the antigen/antibody complexes to Strep-Tactin beads without potential steric restrictions through the bound antibodies. The serum samples used for depletion experiments were obtained from three reconvalescent cattle and one sheep that had previously been infected experimentally with SBV isolate BH80/11–4[23,45].

Prior to depletion, the sera were diluted fivefold or tenfold in 100 mM Tris-HCl pH 8.0, 150 mM NaCl. Aliquots of the diluted sera were then incubated with 30–60 μg of the respective depletion antigens in final volumes of 800 μL each. After 1 h of incubation at 37 °C with over-head rotation, 25 μL of pre-equilibrated Strep-Tactin XT coated magnetic beads (iba Life Science) were added to the mixture and the incubation was continued for 1–2 h. The beads were subsequently removed by magnetic force. If the magnitude of depletion was unsatisfactory after the first round, the procedure was repeated once. Depleted sera were stored at −20 °C until the day of analysis by in-house ELISA or microneutralization assay.

**In-house ELISA for serum depletion analysis**. Medium-binding ELISA plates (Greiner) were coated with 100 ng/well of antigens overnight at 4 °C in 0.1 M carbonate buffer at pH 9.6. The plates were then blocked for 1 h at 37 °C using 5% skimmed milk in PBS. Depleted sera were diluted fivefold in PBS with 0.05% Tween-20 and were incubated on the coated wells for 1 h at 37 °C. HRP-conjugated anti-bovine (Sigma-Aldrich, catalogue number: A5295–1ML) or anti-ovine (Dianova, catalogue number: 313–036–003) secondary antibodies were diluted 1:20,000 or 1:10,000, respectively, and then added for 1 h at room temperature. Following

subsequent addition of TMB substrate (Thermo Scientific), the ELISA readings were taken at a wavelength of 450 nm on a Tecan Infinite F200 instrument. In between each step, the plates were washed three times with PBS/0.05% Tween. Absorbance measurements were normalized to the respective mock-depleted samples. Each sample from cattle was analyzed in duplicates. The samples from the sheep were analyzed in six replicates on the Gc ectodomain as an ELISA antigen, and in four replicates on the respective depletion antigens as ELISA antigens.

**Microneutralization assay for serum depletion analysis**. BHK-21 cells were cultured at 37 °C with 5% $CO_2$ in the MEM supplemented with Earle's salts, nonessential amino acids, and 10% FCS (Cell Culture Collection of Veterinary Medicine, Biobank of the Friedrich-Loeffler-Institut). Microneutralization assays against SBV isolate BH80/11-4 were performed in 96-well microtiter plates using BHK-21 cells[46]. Briefly, for each serum sample twofold dilutions from 1:5 to 1:640 were prepared in a volume of 50 μL per well. Subsequently, 50 μL of a virus suspension containing 100 $TCID_{50}$ of the test virus were added to each well. Serum dilutions mixed with virus suspension were then incubated for 2 h at 37 °C and 5% $CO_2$. After incubation, 100 μL of an appropriate cell suspension was added to each well. Evaluation was performed by assessment of the cytopathic effect, when the cells reached confluence after 3 to 4 days incubation at 37 °C and 5% $CO_2$. All samples were tested in quadruplicates for calculation of neutralizing antibody titers according to Behrens and Kärber[47]. Neutralizing titers of each depleted serum were normalized to the titer of the respective mock-depleted sera. For cattle sera, each sample was tested in three to four independent assays. For the sheep serum, the assay was repeated six times. The results for the three cattle sera or for the one sheep serum were expressed as mean values of all biological and technical replicates for each depletion antigen.

**Mouse blood and serum analysis**. RNA-extraction was performed from 20 μL of EDTA–blood using a KingFisher 96 Flex (Thermo Scientific) and the MagAttract Virus Mini M48 Kit (Qiagen) according to the manufacturer's instructions. RT-qPCR analysis was performed on a BioRad CFX96 Touch real-time PCR detection system (BioRad). An assay detecting an 88 bp fragment within the SBV S segment was applied in combination with the housekeeping gene for beta actin as an internal control (Oligonucleotide sequences are provided in Supplementary Table 5). The reactions were performed using the AgPath-ID One-Step RT-PCR Kit (Applied Biosystems, CA, USA) in a total volume of 25 μL, containing 5 μL of RNA template. For amplification and detection, the following protocol was used: reverse transcription for 10 min at 45 °C, inactivation of reverse transcriptase/activation of Taq polymerase for 10 min at 95 °C, followed by 42 cycles of 15 s at 95 °C, 20 s at 55 °C, and 30 s at 72 °C. An external standard was used for viral RNA quantification[48]. Serum samples obtained at 21 dpi were additionally analyzed with a commercial N protein-based ELISA (ID Screen Schmallenberg virus Competition Multi-Species, ID.vet) following the manufacturer's recommendations.

**Monoclonal antibody cloning and sequencing**. Isotyping of the original monoclonal antibodies 1C11 and 4B6 was performed using the Mouse Monoclonal Antibody Isotyping Test Kit (BioRad). For each hybridoma cell line, total RNA was extracted from one 25-cm² flask using TRIzol reagent (Invitrogen) according to the manufacturer's instructions. Reverse transcription of mRNA was performed using SuperScript III reverse transcriptase (Invitrogen) with an oligo(dT) primer for 1 h at 45 °C. Sets of degenerate primers (Supplementary Table 6) were used to amplify the variable and constant regions of the IgG2a heavy chain (HC) and the variable regions of the light chain (LC). cDNA amplification was performed using GoTaq polymerase (Promega) in a 20 -μL reaction volume according to the manufacturer's instructions. PCR products obtained for HC and LC were cloned into expression plasmids based on pVITRO2-hygro-mcs (Invivogen) yielding in pVITRO-mHC or pVITRO-mLC, respectively. The pVITRO-LC backbone contained a previously inserted constant region of mouse Ig kappa light chain. Multiple clones of each construct were sequenced using pVITRO-specific primers and the BigDye Terminator v1.1 Cycle Seq Kit (Thermo Scientific) on a 3130xl Genetic Analyzer (Applied Biosystems). Sequences were analyzed using IMGT tools[49]. Functionality of the obtained sequences was confirmed by immunoflourescence and ELISA.

**Recombinant IgG2a expression and purification**. FreeStyle 293-F cells (Thermo Fisher) were grown in FreeStyle 293 expression medium (Thermo Fisher) according to the manufacturer's recommendations. The temperature was 37 °C, the shaker speed was 130 rpm, and the $CO_2$ pressure was 8%. Transfection was carried out on freshly passaged cells that had been adjusted to a density of $2 \times 10^6$ cells/mL. Expression plasmids for the heavy chain and the light chain were added to the culture at final concentrations of 1.5 μg/mL each. Polyethylenimine-25K (Polysciences) was added as a transfection agent at a final concentration of 9 μg/mL. Twenty hours after transfection, the culture was supplemented with 3 mg/mL glucose and 3.7 mM valproic acid (Sigma). Culture supernatants were harvested 5–6 days after transfection when cell viability dropped below 70%. Secreted antibodies were bound to a HiTrap 1 mL Protein A HP column (GE Healthcare) that was subsequently washed with 10 mL of 20 mM Tris-Cl, 150 mM NaCl. Antibodies were eluted in 50 mM phosphate/citrate buffer at pH 4.0 and were immediately pH-adjusted with 200 mM Tris-Cl pH 8. The antibodies were further purified by

gel permeation chromatography on a HiLoad Superdex 200 pg column (GE Healthcare) in 20 mM Tris-Cl pH 8.0, 150 mM NaCl.

**Biolayer interferometry**. The affinity of the head–stalk fragment of SBV Gc (aa 465–874) for recombinant full-length antibodies 1C11 and 4B6 was determined by biolayer interferometry on an Octet RED384 instrument (ForteBio). Each antibody was immobilized on anti-mouse IgG Fc Capture (AMC) biosensors (ForteBio) at a concentration of 10 μg/mL. After washing in PBS–BSA (0.2 mg/mL) for 5 min, the loaded sensors were transferred into solutions of different antigen concentrations ranging from 3.13 nM to 100 nM in PBS–BSA (0.2 mg/mL). Antibody-loaded reference sensors were also transferred into PBS–BSA (0.2 mg/mL) without antigen. Association was monitored for 20 min, and data were recorded in duplicates. The reference data were subtracted and steady-state response levels were extrapolated by nonlinear regression. These steady-state response levels were then used to fit a simple kinetic model for estimation of the dissociation constant $K_D$. All data analysis was carried out using the Octet Data Analysis HT software (ForteBio).

**Focus reduction neutralization test**. Vero 76 cells were cultured at 37 °C with 5% $CO_2$ in the MEM supplemented with Earle's salts, nonessential amino acids, and 10% FCS (Cell Culture Collection of Veterinary Medicine, Biobank of the Friedrich-Loeffler-Institut). Serial dilutions of recombinant antibodies were prepared in MEM with 5% FCS and pre-incubated with ~300 FFU of SBV isolate BH80/11–4 in a total volume of 400 μL for 2 h at 37 °C. The antibody–virus mixtures and virus controls (virus dilutions without antibodies) were then added onto Vero 76 cell monolayers in 12-well tissue culture plates (Corning). After incubation for 2 h at 37 °C, the inoculum was removed and the cells were washed once with PBS. The cells were then overlaid with 0.75% methyl cellulose in MEM supplemented with 1% FCS and antibiotics. After incubation for 72 h at 37 °C, the cells were fixed with 4% paraformaldehyde for 20–30 min at room temperature and were permeabilized for 5 min with 0.5% Triton X-100 in PBS. Staining for immunofluorescence analysis was performed using the original hybridoma supernatant of monoclonal antibody 1C11, specific for SBV Gc, at a dilution of 1:100 in PBS + 0.05% Tween. An anti-mouse Alexa-488-labeled secondary antibody was diluted 1:100 and used for detection (Molecular Probes, Invitrogen, catalogue number: A-11017). The total number of foci in each well was counted using a Nikon Eclipse Ti-U inverted microscope with the NIS-Elements Imaging Basic Research software (Nikon). Assays were performed in duplicates. Nonlinear regressions with a Langmuir isotherm were computed using the pro Fit software (QuantumSoft). $ND_{50}$ values were interpolated from the regressions.

**Position-specific sequence similarity plots**. Sequence similarity data in Figs. 1a and 7, and Supplementray Fig. 3d were calculated using the ESPript 3.0 web server[50]. The values displayed in Fig. 1a were averaged over a window of 15 residues to smoothen the plot. Multiple sequence alignments of the following GenBank entries were created using Clustal Omega[51].

For Fig. 1a, Schmallenberg virus (CCF55030), Sathuperi virus (BAM15761), Douglas virus (BAM15767), Shuni virus (AQM56673), Aino virus (CCG93468), Simbu virus (CCG93496), Shamonda virus (BAM15764), Sango virus (CCG93484), Peaton virus (BAJ16510), Akabane virus (BAV17033), Sabo virus (CCG93480), Tinaroo virus (BAF91637), Perdoes virus (AJT39462), Jatobal virus (AFI24667), Utive virus (AHY22346), Utinga virus (AHY22345), Oropouche virus (AJT39470), Iquitos virus (AIK67306), Madre de Dios virus (AHY22342), Facey's Paddock virus (AHY22339), Buttonwillow virus (AHY22347), Oya virus (ARF07018), Ingwavuma virus (AHY22340), Mermet virus (AHY22344), Manzanilla virus (AHY22343), Leanyer virus (AEA02983), Bahig virus (AKO90171), Matruh virus (AKO90184), Tete virus (AKO90180), I612045 virus (AED98379), Brazoran virus (AGS94385), Oriboca virus (AGW82131), Marituba virus (AGW82142), Itaya virus (AJZ73168), Madrid virus (AGW82139), Bruconha virus (AKB96238), Caraparu virus (AGW82158), Mirim virus (APM83099), Bellavista virus (ANI69985), Enseada virus (AMR98953), Bimiti virus (AKO90172), Cati virus (AKO90173), Ananindeua virus (APM83096), Guama virus (AKO90175), Mahogany Hammock virus (ALP92134), Moju virus (AKO90178), Capim virus (ALP92389), Guajara virus (AKO90174), Bunyamwera virus (AAA42777), Ilesha virus (AGU99596), Anadyr virus (ALN13304), Ngari virus (AFY23375), Batai virus (AGM39999), Tensaw virus (ACV95627), Cache Valley virus (AGI03947), Maguari virus (AAQ23639), Northway virus (ABV68911), Abbey Lake virus (AIA08881), Germiston virus (AAA42778), Main Drain virus (ABV68910), Potosi virus (ABV68912), Kairi virus (ALC78365), Cachoeira Porteira virus (AEZ35284), Sororoca virus (AEZ35265), Iaco virus (AEZ35259), Anhembi virus (AEZ35256), Macaua virus (AEZ35262), Taiassui virus (AEZ35268), Wyeomyia virus (AGA54137), Guaroa virus (AAR28442), La Crosse virus (ABQ12634), Snowshoe hare virus (ABX47014), Chatanga virus (AEK05548), California encephalitis virus (APA29014), Tahyna virus (ACV04452), Lumbo virus (AAD53040), San Angelo virus (APA28990), South River virus (AFI56181), Inkoo virus (AAB93841), Jamestown Canyon virus (ADK36673), Keystone virus (ALD52503), Serra do Navio virus (APA28986), Melao virus (APA28994), Trivittatus virus (ALI93834), Bwamba virus (AIN37029), Pongola virus (AIN37027), Calchaqui virus (AIS74642), Alajuela virus (AIS74643), Gamboa virus (AIS74641), Umbre virus (AKO90182), Witwatersrant virus (AKO90183), Kaeng Khoi virus (AIN37035), Mojui dos Campos virus (AIN37034),

Nyando virus (AIN37030), Lukuni virus (AKO90177), Tataguine virus (AKO90179), Maprik virus (AJD77606), Salt Ash virus (AGX00985), Mapputta virus (AKO90185), and Buffalo Creek virus (AJD77609).

For Supplementary Fig. 3d, Bunyamwera serogroup: Bunyamwera virus (AAA42777), Ilesha virus (AGU99596), Anadyr virus (ALN13304), Ngari virus (AFY23375), Batai virus (AGM39999), Tensaw virus (ACV95627), Cache Valley virus (AGI03947), Maguari virus (AAQ23639), Northway virus (ABV68911), Abbey Lake virus (AIA08881), Germiston virus (AAA42778), Main Drain virus (ABV68910), Potosi virus (ABV68912), Kairi virus (ALC78365), Cachoeira Porteira virus (AEZ35284), Sororoca virus (AEZ35265), Iaco virus (AEZ35259), Anhembi virus (AEZ35256), Macaua virus (AEZ35262), Taiassui virus (AEZ35268), Wyeomyia virus (AGA54137), and Guaroa virus (AAR28442).

For Supplementary Fig. 3d, California serogroup: La Crosse virus (ABQ12634), Snowshoe hare virus (ABX47014), Chatanga virus (AEK05548), California encephalitis virus (APA29014), Tahyna virus (ACV04452), Lumbo virus (AAD53040), San Angelo virus (APA28990), South River virus (AFI56181), Inkoo virus (AAB93841), Jamestown Canyon virus (ADK36673), Keystone virus (ALD52503), Serra do Navio virus (APA28986), Melao virus (APA28994), and Trivittatus virus (ALI93834).

For Supplementary Fig. 3d, Simbu serogroup viruses with OROV disulfide pattern: Oropouche virus (AJT39470), Iquitos virus (AIK67306), Madre de Dios virus (AHY22342), Facey's Paddock virus (AHY22339), Buttonwillow virus (AHY22347), Oya virus (ARF07018), Ingwavuma virus (AHY22340), Mermet virus (AHY22344), Manzanilla virus (AHY22343), and Leanyer virus (AEA02983).

For Supplementary Fig. 3d and Fig. 7a, Simbu serogroup viruses with SBV disulfide pattern: Schmallenberg virus BH80/11-4 (CCF55030), Sathuperi virus KSB-2/C/08 (BAM15763), Douglas virus CSIRO150 (BAM15767), Shuni virus Ib An 10107 (CCH15004), Aino virus B7974 (BAJ16505), and Aino virus KS-1/P/98 (BAJ16503).

For Fig. 7b, SBV BH80/11-4 (CCF55030), SBV BH619/12-1 (AGC93534), SBV BH635/12-2 (AGC93535), SBV BH652/12-1 (AGC93536), SBV D495/12-1 (AKA63345), SBV F6 (AGU16236), SBV 79.4 (AIP98327), SBV 91.1 (AIP98328), SBV 96.1 (AIP98329), SBV 100.3 (AIP98330), SBV 102.2 (AIP98331), SBV 175.2 (AIP98332), and SBV 200.2 (AIP98333).

For Fig. 7c, SBV HL1 (AGU16232), SBV 1916 (LC309139), SBV 1892 (LC309138), SBV 1871 (LC309137), SBV 1844 (LC309136), SBV 1833 (LC309135), SBV 1583 (LC309134), SBV 941 (LC309133), SBV 901 (LC309132), SBV 242 (LC309131), SBV 182 (LC309130), SBV 174 (LC309129), SBV 167 (LC309128), SBV 138 (LC309127), SBV 239 (LC309126), SBV 225 (LC309125), SBV 777 (LC309124), SBV BH336/12-3 (AGC93533), SBV BH336/12-1 (AGC93532), SBV BH250/12-2 (AGC93531), SBV BH248/12-1 (AGC93530), SBV BH237/12-4 (AGC93529), SBV BH233/12-1 (AGC93528), SBV BH231/12-1 (AGC93527), SBV BH200/12-2 (AGC93526), SBV BH199/12-5 (AGC93525), SBV BH198/12-5 (AGC93524), SBV BH197/12-3 (AGC93523), SBV BH174/12-2 (AGC93522), SBV BH148/12-9 (AGC93521), SBV BH127/12-16 (AGC93520), SBV BH59/12-8 (AGC93518), SBV BH37/12-2 (AGC93517), SBV BH28/12-5 (AGC93516), SBV BH03/12-3 (AGC93515), SBV BH02/12-1 (AGC93514), SBV Na2-CNS (AGD94646), and SBV Na1-CNS (AGD94642).

**Quantification and statistical analysis**. Data are represented as means ± s.d. unless otherwise indicated. Following tests for normality, pairwise statistical comparisons were performed using the Mann–Whitney rank sum test with the software SigmaPlot 11 (Systat Software).

**Reporting summary**. Further information on experimental design is available in the Nature Research Reporting Summary linked to this article.

## Data availability

Atomic coordinates, structure factor amplitudes, and the respective protein sequences have been deposited in the Protein Data Bank (PDB). Accession numbers are 6H3S for SBV Gc(465–874), 6H3T for SBV Gc(465–702)/scFv 1C11, 6H3U for SBV Gc(465–702)/scFv 4B6, 6H3V for BUNV Gc(478–721), 6H3W for LACV Gc(477–722), and 6H3X for OROV Gc(482–702). The source data underlying Figs. 3, 4, 5a, 5b, and 6 are provided as a Source Data file. A reporting summary for this Article is available as a Supplementary Information file. All other data supporting the findings of this study are available from the corresponding authors upon request.

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

## Acknowledgements

This research was performed as part of the Zoonoses Anticipation and Preparedness Initiative (ZAPI project; IMI Grant Agreement n°115760), with the assistance and financial support of IMI and the European Commission, and in kind contributions from EFPIA partners. F.A.R. also received funding by the Région Ile de France (Domaine d'intêret majeur Innovative technologies for life sciences, DIM 1HEALTH). Additional funding was provided by Institut Pasteur, the CNRS and the GIS IBiSA (Infrastructures en biologie santé et agronomie). J.H. received the Pasteur-Cantarini fellowship for 24 months, and was later supported by the DIM 1HEALTH grant. We thank Patrick England from the Molecular Biophysics facility and Fabrice Agou from the Chemogenomic and Biological Screening platform at Institut Pasteur and the staff of synchrotron beamlines PX1 and PX2 at SOLEIL (St. Aubin, France) and ID29 and ID30B at the ESRF (Grenoble, France) for help during data collection. We thank Xiaohong Shi and Richard M. Elliott from Glasgow, UK, for the BUNV Gc gene.

## Author contributions

F.A.R., J.H., M.B., and A.A. designed the experiments. J.H. and A.A. produced the recombinant proteins. J.H. performed the biophysical and crystallographic analysis with contributions by A.H. and P.G.-C. E.B. provided hybridoma cells. A.A. performed the neutralization assays and serum depletion analyses, and together with S.R cloned the mAbs. A.A. and K.W. performed the animal experiments. J.H. and F.A.R. wrote the manuscript with contributions by A.A. and M.B.

## Additional information

**Competing interests:** The authors declare no competing interests.

