## [Peer Review File · Nature Communications]

Reviewers' comments:

Reviewer #1 (Remarks to the Author):

Hellert et al. report X-ray crystal structures of parts of Orthobunyavirus spike. They show that part of the spike comprises an N-terminal extension of the fusion protein (Gc) and contains two domains; a head and a stalk. They report crystal structures of the head and stalk for Schmallenberg virus and of the head for Bunyamwera (BUNV), Lacrosse, and Oropouche (OROV) viruses. They generate a model for the spike by docking their coordinates into a previously published low-resolution EM tomography map of BUNV. They identify the head as a target for protective antibodies and report crystal structures of neutralizing antibody fragments bound to the head domain. They further show that immunizing mice with recombinant protein comprising the head and the stalk domains yields sterilizing immunity in some mice.

The epitope for the protective neutralizing antibodies they describe, however, would not be exposed on the virion surface for the glycoprotein organization they are able to model using the EM maps (these lay at the trimer interface). The authors make the plausible argument that conformational flexibility of the spike could account for this inconsistency, which is in line with the suspected structural plasticity of the envelope proteins for a number of viruses.

The work is thoroughly executed, and the manuscript is well written. The work is also important and of broad interest to the field.

Some modifications are recommended to better represent/show the epitopes of neutralizing antibodies on the modeled spike and to make some of the figures and conclusions clearer.

Major:

The authors should add a figure (to the main or supplemental text) in which the scFvs for 4B6 and 1C11 are docked onto their model of the head and stalk domains as displayed in Figure 2D. This would allow better visualization of the predicted antibody binding sites in the context of their spike

model and bring home the point that the epitopes are not available for antibody binding in the modeled conformation.

The antibody-glycoprotein complex structures raise some important questions that should be addressed in the discussion: how are the antibodies blocking viral entry, could they interfere with receptor binding, or block conformational changes required for entry?

The authors should discuss in more detail whether they believe the “open trimer” organization in Figure 2C, obtained by crystallizing the head domain of OROV, represents a distinct, physiologically relevant conformational state that could be adopted by the spike, or an irrelevant organization?

The authors could comment on whether the OROV “open trimer” as crystallized would fit the BUNV EM map, and whether the 4B6/1C11 epitopes would be more available on this organization.

In terms of pH sensitivity of the proteins, in addition to the pH dependent charge the authors alluded to, they should discuss whether there are obvious histidine residues at the interface that could play in a role.

Minor comments:

Figure 3a.

- Additional numbering (e.g. 50% mark) should be added to the y-axis to make the panel more easily interpretable.
- The authors should also add statistics to Figure 3a (e.g. is the difference they observed between the core and head-stalk constructs statistically significant, or the interpretation meant to be more qualitative)?
- What OD was measured for mock depleted sera with Gc ectodomain and was the signal robust? As the data are normalized, including this number in the legend would be useful.

Figure 5d should include a rotated view of the model so that the contacts being made by 4B6 scFv could be more readily visualized.

Line 42: SBV should be spelled out as Schmallenberg virus (or the abbreviation should be listed on line 34 in the abstract)

Line 211: “We thus confirm that the well-exposed Gc head domain is the major target of neutralizing antibodies not only in mice immunized with infectious virus, but also in native host animals after acute primary infection with SBV”. This part of the text comes before any data on mice immunization studies are presented; the mice experiments should be mentioned later in the text.

Reviewer #2 (Remarks to the Author):

Orthobunyaviruses constitute a genetically diverse group of important human and animal pathogens. Although electron microscopy analysis of Bunyamwera virus, the type virus of the genus, has revealed a tripodal organization of the glycoprotein lattice displayed on the viral envelope surface, there has been a paucity of high resolution structural information about the individual Gn/Gc glycoprotein components that make up the spike complex. In this well carried out study, Hellert, Aebischer, and colleagues report high resolution crystal structures of the Gc N-terminal region for several orthobunyaviruses. This analysis shows that the Gc is formed of a helical head and a multi-domain β -stranded stalk region. Interestingly, several Gc glycoproteins crystallized as trimers and form an organization matching that observed in a previously reported EM reconstruction.

In a series of integrative analyses, the authors also show that the N-terminal head region of the Gc is the primary target of the antibody-mediated immune response, where recombinant SBV Gc elicits a protective antibody response and previously reported neutralizing anti-SBV mAbs prevent against infection. Finally, structural analysis of Fab-Gc complexes reveal neutralizing epitopes targeted by the immune response. The data appear sound, the manuscript was a pleasure to read, and this work provides much needed information about the molecular architecture of these important pathogens.

1. The authors clearly demonstrate OBV Gc head domain as an immunodominant site targeted by the antibody response. (a) Can the authors comment on the mode of action for antibodies directed to this region of the virus? (b) Furthermore, the importance of the nAb-Gc structures is not completely obvious. Do the structures provide any insights into mechanism(s) of action?

2. Mankouri, Barr, and co-workers have shown that potassium is important for host cell entry and glycoprotein structure of OBVs and other bunyaviruses (Hover et al. PLoS Path, 2018; Hover et al.,

JBC, 2016; Punch et al., JBC 2018). Can the authors comment on whether the differential presence of a cation may have an effect on the observed (trimeric or non-trimeric) conformation or functional state of the Gc?

3. Line 385: The authors hypothesize that shielding of the C-terminal (fusogenic) region of the Gc by the N-terminal domain “allows its optimization for function”. The authors should be more explicit. Upon superficial assessment, it appears that the presence of the N-terminus sterically impedes fusogenic rearrangements.

4. Inspection of Fig. 1 reveals that most of the crystallized Gc glycoproteins present well-ordered glycan chains. Were these chains stabilized by the local protein environment or crystal packing?

5. Can the authors comment on whether the pH environment influences the oligomeric state of the N-terminal Gc (both in solution and in the crystallization condition)? Furthermore, it is not obvious what pH was used for the SEC-MALS experiments (Fig. S1a).

6. In addition to the crystallized constructs, the stalk and core ectodomains were used for the antibody depletion experiments. A brief description of the solution state of the stalk/core ectodomains would help to assess whether these regions formed native conformations, which would be recognized by the antibody response.

7. The monomeric N-terminal region of the OBV Gc is shown to be an effective immunogen. As the authors show that this region of the Gc is likely to be presented as a homotrimeric complex on the mature virion surface, would they expect that a stabilized trimer would prove to be an improved immunogen for SBV or other orthobunyaviruses (e.g. by analogy to studies of trimeric HIV env)?

8. References: (a) Line 59: When describing the Bunyavirus M segment, the statement that the Gc glycoprotein is a class-II membrane fusion protein appears somewhat misrepresentative, as it has only been formally shown for a small subset of families within the Bunyavirinae. (b) Line 62: Similarly, the reference to Gn and Gc associating co-translationally is specific to a review on OBVs (ref 13), while the text appears to be generalizing all bunyaviruses.

9. The sequence conservation bars shown in Figs. 7 and S1 are qualitatively annotated from ‘+’ to ‘-’. These values should be quantified as it is difficult to interpret the range of sequence diversification from these relative values.

Reviewer #3 (Remarks to the Author):

Bunyaviruses are a large and diverse group of viruses, a number of which are human pathogens and perhaps a larger number that have the potential of becoming of significant clinical concern. Among medically important viruses, the Bunya's have lagged behind others with regards to structural studies. Recently, some very nice cryo EM studies have provided quite a bit of insight into the structures of Bunyavirus spike proteins. In this study, the Rey lab has moved things forward by providing detailed crystallographic structure information on several Orthobunyavirus spike proteins (large fragments thereof). Putting this work together with the earlier cryo EM studies gives a very nice view of these proteins. The authors, using this structural information, made predictions on where neutralizing antibodies are directed, and what regions of the spike proteins might make good immunogens. The later half of this study showed these predictions to be true, and while necessarily somewhat superficial at this early stage, do a nice job of using the crystal structures and suggest new approaches for vaccine development. As is typical of this lab, the paper is well put together and to the point. Very nice job!

We thank the reviewers for their constructive comments and suggestions. E have now implemented them, and feel that our manuscript has been improved by adding the required clarifications. We provide a point-by-pont answer below:

Reviewer #1 (Remarks to the Author):

Hellert et al. report X-ray crystal structures of parts of Orthobunyavirus spike. They show that part of the spike comprises an N-terminal extension of the fusion protein (Gc) and contains two domains; a head and a stalk. They report crystal structures of the head and stalk for Schmallenberg virus and of the head for Bunyamwera (BUNV), Lacrosse, and Oropouche (OROV) viruses. They generate a model for the spike by docking their coordinates into a previously published low-resolution EM tomography map of BUNV. They identify the head as a target for protective antibodies and report crystal structures of neutralizing antibody fragments bound to the head domain. They further show that immunizing mice with recombinant protein comprising the head and the stalk domains yields sterilizing immunity in some mice.

The epitope for the protective neutralizing antibodies they describe, however, would not be exposed on the virion surface for the glycoprotein organization they are able to model using the EM maps (these lay at the trimer interface). The authors make the plausible argument that conformational flexibility of the spike could account for this inconsistency, which is in line with the suspected structural plasticity of the envelope proteins for a number of viruses.

The work is thoroughly executed, and the manuscript is well written. The work is also important and of broad interest to the field.

Some modifications are recommended to better represent/show the epitopes of neutralizing antibodies on the modeled spike and to make some of the figures and conclusions clearer.

Major:

The authors should add a figure (to the main or supplemental text) in which the scFvs for 4B6 and 1C11 are docked onto their model of the head and stalk domains as displayed in Figure 2D. This would allow better visualization of the predicted antibody binding sites in the context of their spike model and bring home the point that the epitopes are not available for antibody binding in the modeled conformation.

We appreciate the reviewer's suggestion. We have now modified Figure 5 to add scFvs in the context of the spike, explicitly showing that their binding site is occluded.

The antibody-glycoprotein complex structures raise some important questions that should be addressed in the discussion: how are the antibodies blocking viral entry, could they interfere with receptor binding, or block conformational changes required for entry?

Following this comment, we appended the following statement to our discussion of the conformational flexibility of the spike (lines 307-310): *"The neutralization mechanism of our antibodies is therefore likely based on sterical hindrance of host cell attachment or of membrane fusion, while a potential destabilization of the envelope glycoprotein lattice may have a contributing effect."*

The authors should discuss in more detail whether they believe the “open trimer” organization in Figure 2C, obtained by crystallizing the head domain of OROV, represents a distinct, physiologically relevant conformational state that could be adopted by the spike, or an irrelevant organization?

This is a good point. The observed organization of the “open trimer” in the crystals is exclusively stabilized by crystal-packing contacts, and on the viral envelope there would be no equivalent restraints to maintain the subunits in this specific arrangement. To further illustrate this aspect, we added the new Supplementary Figure 4 describing the crystal packing of the OROV Gc head domain. However, as already discussed in context of our monoclonal antibodies, it appears likely that open arrangements are also adopted at least transiently within the spike.

The authors could comment on whether the OROV “open trimer” as crystallized would fit the BUNV EM map, and whether the 4B6/1C11 epitopes would be more available on this organization.

In line with our comment above, we feel that a discussion of this question is not suited for the main text of the manuscript. To answer the question, the ‘open trimer’ fits less well into the head region of the BUNV spike than the closed trimer, but the epitopes of both our monoclonal antibodies would still be sterically inaccessible, as the three monomers are not far enough apart from each other.

In terms of pH sensitivity of the proteins, in addition to the pH dependent charge the authors alluded to, they should discuss whether there are obvious histidine residues at the interface that could play in a role.

Following this good point, we implemented a new Supplementary Figure 3b showing the chemical environment of the histidine side chains at the trimer interfaces of BUNV and LACV. A modified statement in the text now reads (lines 167-172): *“The dissociation of the trimers at low pH is likely triggered by protonation of residue H590 in BUNV and in LACV, which would lead to electrostatic repulsion with positively charged side chains on the neighboring protomer (Supplementary Fig. 3b-c). Although the position of histidine residues at the interface is not strictly conserved, there is at least one histidine with presumably similar function present at each of the inferred trimerization sites of SBV and OROV (Fig. 1d).”*

Minor comments:

Figure 3a.

- Additional numbering (e.g. 50% mark) should be added to the y-axis to make the panel more easily interpretable.

We thank the reviewer for this comment. We now implemented this suggestion in the new Fig. 3a.

- The authors should also add statistics to Figure 3a (e.g. is the difference they observed between the core and head-stalk constructs statistically significant, or the interpretation meant to be more qualitative)?

The ELISA data had been recorded in duplicates. We now repeated the ELISA for the sheep samples with six replicates each to reach statistical significance with $P < 0.01$.

- What OD was measured for mock depleted sera with Gc ectodomain and was the signal robust? As the data are normalized, including this number in the legend would be useful.

We now added a remark in the legend of Figure 3, stating (Lines 1036-1038): *“Bars represent mean percentages of mock-depleted sera, the OD values of which ranged robustly between 3.2 and 3.6 for reactivity with the Gc ectodomain.”* The raw data are also available in our Source Data table.

Figure 5d should include a rotated view of the model so that the contacts being made by 4B6 scFv could be more readily visualized.

We took this suggestion into account during our revision of Figure 5.

Line 42: SBV should be spelled out as Schmallenberg virus (or the abbreviation should be listed on line 34 in the abstract)

As rightly suggested, we now spelled out “Schmallenberg virus” at its first appearance in the introduction.

Line 211: “We thus confirm that the well-exposed Gc head domain is the major target of neutralizing antibodies not only in mice immunized with infectious virus, but also in native host animals after acute primary infection with SBV”. This part of the text comes before any data on mice immunization studies are presented; the mice experiments should be mentioned later in the text.

We now rephrased this sentence and included for clarity the respective citations of Kingsford et al. (1983) and Roman-Sosa et al. (2016), which were already cited at the beginning of this chapter. The sentence now reads (lines 222-225): *“Thus, in agreement with the previous findings from mice immunized with infectious virus, we show that the well-exposed Gc head domain is also the major target of neutralizing antibodies in native host animals after acute primary infection with SBV.”*

Reviewer #2 (Remarks to the Author):

Orthobunyaviruses constitute a genetically diverse group of important human and animal pathogens. Although electron microscopy analysis of Bunyamwera virus, the type virus of the genus, has revealed a tripodal organization of the glycoprotein lattice displayed on the viral envelope surface, there has been a paucity of high resolution structural information about the individual Gn/Gc glycoprotein components that makeup the spike complex. In this well carried out study, Hellert, Aebischer, and colleagues report high resolution crystal structures of the Gc N-terminal region for several orthobunyaviruses. This analysis shows that the Gc is formed of a helical head and a multi-domain b-stranded stalk region. Interestingly, several Gc glycoproteins crystalized as trimers and form an organization matching that observed in a previously reported EM reconstruction.

In a series of integrative analyses, the authors also show that the N-terminal head region of the Gc is the primary target of the antibody-mediated immune response, where recombinant SBV Gc elicits a protective antibody response and previously reported neutralizing anti-SBV mAbs prevent against infection. Finally, structural analysis of Fab-Gc complexes reveal neutralizing epitopes targeted by the immune response. The data appear sound, the manuscript was a pleasure to read, and this work provides much needed information about the molecular architecture of these important pathogens.

1. The authors clearly demonstrate OBV Gc head domain as an immunodominant site targeted by the antibody response. (a) Can the authors comment on the mode of action for antibodies directed to this region of the virus? (b) Furthermore, the importance of the nAb-Gc structures is not completely obvious. Do the structures provide any insights into mechanism(s) of action?

Our crystal structures show that the antibodies bind to epitopes that would be inaccessible in the closed spike, supporting the notion that the spike transiently opens under physiological conditions. As already suggested by the first reviewer, we addressed this point with a revision of Figure 5 and a brief statement in the text saying (lines 307-310): *“The neutralization mechanism of our antibodies is therefore likely based on sterical hindrance of host cell attachment or of membrane fusion, while a potential destabilization of the envelope glycoprotein lattice may contribute to the neutralization effect.”*

2. Mankouri, Barr, and co-workers have shown that potassium is important for host cell entry and glycoprotein structure of OBVs and other bunyaviruses (Hover et al. PLoS Path, 2018; Hover et al., JBC, 2016; Punch et al., JBC 2018). Can the authors comment on whether the differential presence of a cation may have an effect on the observed (trimeric or non-trimeric) conformation or functional state of the Gc?

These recent reports about the role of potassium are interesting indeed. However, none of the crystals from our current study were obtained in the presence of potassium, so to this end we have no information on any specific effect of this cation on the Orthobunyavirus spike structure. Following the reviewer’s remark, we now performed additional SEC-MALS experiments, displayed in Supplementary Figure 2a demonstrating that substitution of 150 mM NaCl by 150 mM KCl in the running buffer does not induce oligomerization of the Gc head domains in solution. We now also cite the study from Hover et al., PLoS Pathogens (2018) in the respective section of the main text, which now reads (lines 140-142): *“The recombinant head domains of Gc from all four viruses behaved as monomers in solution, both at pH 8.0 and at pH 5.5. The presence of 150 mM potassium, which was recently shown to promote BUNV infection in host endosomes²⁰, did not affect the oligomeric state (Supplementary Fig. 2a).”*

A previous study by Guardado-Calvo et al., PLoS Pathogens (2016) from our lab showed that potassium can affect the local hydrogen bond network near the fusion loops of the Hantavirus Gc protein, which we consider a structural and functional homolog of the C-terminal core domain of Orthobunyavirus Gc.

3. Line 385: The authors hypothesize that shielding of the C-terminal (fusogenic) region of the Gc by the N-terminal domain “allows its optimization for function”. The authors should be more explicit. Upon superficial assessment, it appears that the presence of the N-terminus sterically impedes fusogenic rearrangements.

We apologize - this sentence was indeed not very clear. We meant to say: *“We hypothesize that shielding of the fusion machinery from the humoral adaptive immune system allows its evolutionary optimization for function by limiting antigenic drift, whereas the burden of escaping the immune pressure lies primarily on the projecting variable domains.” (lines 398-401 in the revised manuscript).*

4. Inspection of Fig. 1 reveals that most of the crystallized Gc glycoproteins present well-ordered glycan chains. Were these chains stabilized by the local protein environment or crystal packing?

The glycan at position N493 is indeed stabilized by the local protein environment. We now provide a new Supplementary Figure 1 showing the respective electron density maps in the

structures with scFv 1C11 and scFv 4B6. The corresponding main text now reads (lines 102-104): “Of note, the glycan chain at position N493, which was earlier shown to be essential for protein folding and secretion, is well ordered through stabilization by the local protein environment.” In contrast, the ordered glycan in the LACV structure (Fig. 2b) is stabilized by crystal contacts.

5. Can the authors comment on whether the pH environment influences the oligomeric state of the N-terminal Gc (both in solution and in the crystallization condition)? Furthermore, it is not obvious what pH was used for the SEC-MALS experiments (Fig. S1a).

Our original SEC-MALS experiments were performed in 20 mM Tris-Cl pH8.0, 150 mM NaCl. We now also performed equivalent experiments in 20 mM MES pH5.5, 150 mM NaCl, which are presented in Supplementary Figure 2a. They show that in solution the head domain is monomeric irrespective of the buffer. Regarding the crystallization conditions, we now added the following statement to the text (lines 176-177)*: “[...] all the three head domain structures were obtained in the same pH range between pH 7.5 and pH 8.5.”

6. In addition to the crystallized constructs, the stalk and core ectodomains were used for the antibody depletion experiments. A brief description of the solution state of the stalk/core ectodomains would help to assess whether these regions formed native conformations, which would be recognized by the antibody response.

We now added the Supplementary Figure 2b, showing that all constructs used for serum depletion were monomeric in solution. The main text correspondingly reads (lines 201-202): “All four constructs behaved correctly and eluted as monomers in solution (Supplementary Fig. 2b), indicating that their native fold was preserved.”

[Redacted]

8. References: (a) Line 59: When describing the Bunyavirus M segment, the statement that the Gc glycoprotein is a class-II membrane fusion protein appears somewhat misrepresentative, as it has only been formally shown for a small subset of families within the Bunyavirinae. (b) Line 62: Similarly, the reference to Gn and Gc associating co-translationally is specific to a review on OBVs (ref 13), while the text appears to be generalizing all bunyaviruses.

In our revised manuscript, these statements have now been changed to apply only to Orthobunyaviruses and no longer to the entire order of the Bunyavirales (lines 60-66): “The OBV M segment encodes two glycoproteins, Gn and Gc, derived from a single polyprotein precursor, with Gn encoded upstream of Gc. A non-structural protein denoted NSm is encoded in between Gn and Gc. Glycoprotein Gc is predicted to contain a class II membrane fusion protein fold, structurally homologous to the fusion glycoproteins of the flaviviruses and the alphaviruses. Gn and Gc associate cotranslationally in the ER lumen of infected cells, and the resulting heterodimer is transported to the Golgi apparatus where new virions bud.”

9. The sequence conservation bars shown in Figs. 7 and S1 are qualitatively annotated from ‘+’ to ‘-’. These values should be quantified as it is difficult to interpret the range of sequence diversification from these relative values.

The *ESPrpt* web server that we used to generate our conservation plots does not provide quantitative values. To account for this shortcoming, we now clearly marked the coloring as

“relative” in each relevant figure legend. In order to still provide the reader with precise data, the underlying formatted alignments are available as Supplementary Data 1 and 2.

Reviewer #3 (Remarks to the Author):

Bunyaviruses are a large a diverse group of viruses, a number of which are human pathogens and perhaps a larger number that have the potential of becoming of significant clinical concern. Among medically important viruses, the Bunya's have lagged behind others with regards to structural studies. Recently, some very nice cryo EM studies have provided quite a bit of insight into the structures of Bunyavirus spike proteins. In this study, the Rey lab has moved things forward by providing detailed crystallographic structure information on several Orthobunyavirus spike proteins (large fragments thereof). Putting this work together with the earlier cryo EM studies gives a very nice view of these proteins. The authors, using this structural information, made predictions on where neutralizing antibodies are directed, and what regions of the spike proteins might make good immunogens. The later half of this study showed these predictions to be true, and while necessarily somewhat superficial at this early stage, do a nice job of using the crystal structures and suggest new approaches for vaccine development. As is typical of this lab, the paper is well put together and to the point. Very nice job!

We thank this reviewer for his/her positive feedback.

**** See Nature Research's author and referees' website at www.nature.com/authors for information about policies, services and author benefits**

REVIEWERS' COMMENTS:

Reviewer #1 (Remarks to the Author):

The authors have answered my comments satisfactorily and I have no further concerns.